# Equal Opportunity of Coverage in Fair Regression

**Fangxin Wang**
University of Illinois Chicago
Chicago, USA
fwang51@uic.edu

**Lu Cheng**
University of Illinois Chicago
Chicago, USA
lucheng@uic.edu

**Ruocheng Guo**
ByteDance Research
London, UK
rguo.asu@gmail.com

**Kay Liu**
University of Illinois Chicago
Chicago, USA
zliu234@uic.edu

**Philip S. Yu**
University of Illinois Chicago
Chicago, USA
psyu@uic.edu

## Abstract

We study fair machine learning (ML) under predictive uncertainty to enable reliable and trustworthy decision-making. The seminal work of "equalized coverage" proposed an uncertainty-aware fairness notion. However, it does not guarantee equal coverage rates across more fine-grained groups (e.g., low-income females) conditioning on the true label and is biased in the assessment of uncertainty. To tackle these limitations, we propose a new uncertainty-aware fairness – Equal Opportunity of Coverage (EOC) – that aims to achieve two properties: (1) coverage rates for different groups with similar outcomes are close, and (2) the coverage rate for the entire population remains at a predetermined level. Further, the prediction intervals should be narrow to be informative. We propose Binned Fair Quantile Regression (BFQR), a distribution-free post-processing method to improve EOC with reasonable width for *any* trained ML models. It first calibrates a hold-out set to bound deviation from EOC, then leverages conformal prediction to maintain EOC on a test set, meanwhile optimizing prediction interval width. Experimental results demonstrate the effectiveness of our method in improving EOC. Our code is publicly available at https://github.com/fangxin-wang/bfqr.

## 1 Introduction

Machine Learning (ML) can bring bias and discrimination even with good intentions [1, 2, 3, 4, 5, 6]. Fair ML has been developed to counteract unfairness, but the practical use of fair ML models is limited by predictive uncertainty. Predictive uncertainty is the extent to which ML can confidently predict the future. Over- or under-confidence can cause an ML model to be unaware of its own knowledge gaps and make inaccurate predictions [7, 8, 9]. This can lead to unfairness in decision-making. To address this, we can produce predicted intervals for each sample and incorporate uncertainty into fairness to make decisions more reliable and trustworthy.

The idea of "equalized coverage" – an uncertainty-aware notion of demographic parity [10] – was introduced in a study [11] as a way to ensure that every group receives the same level of prediction certainty. It works by generating prediction intervals that cover the true label $Y$ with a specified probability (e.g., 90%), while also reflecting uncertainty through interval width. However, even with this approach, there are still disparities in coverage rates across groups when conditioning on $Y$. For example, we observe from the empirical results (Fig. 1) for *Adult* dataset [4] that low-income women are less likely to be covered than men in the same income bracket, and high-income men are less likely to be predicted to earn as much as high-income women. Consequently, the widths of prediction intervals for different groups, as an indicator of the uncertainty, are not comparable under

37th Conference on Neural Information Processing Systems (NeurIPS 2023).

different coverage rates. These disparities can lead to unfair risk assessment for domains like bank loans and taxation. Further, equalized coverage may sacrifice the efficiency of uncertainty estimation for ensuring coverage rate as it produces wider prediction intervals for the group it intends to protect (See Section 5).

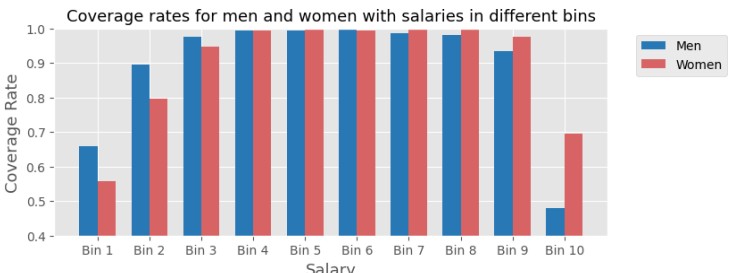

Figure 1: Evaluating equalized coverage [11] on the *Adult* dataset. The protected attribute is gender. The test data is partitioned into 10 equal-mass bins based on the ascending order of salaries. Significant variations in the coverage rates can be observed among different groups within the head and tail bins.

To address the limitations of equalized coverage, we propose a novel uncertainty-based fairness notion *Equal Opportunity of Coverage (EOC)*, extending from the standard fairness notion of equal opportunity [12]. EOC aims to achieve two properties: (1) similar coverage rates for different groups (e.g., female and male) with similar outcomes $Y$ (e.g., salaries), (2) achieve a desired level of coverage rate (e.g., 90%) for the entire population. Ideally, to provide informative predictions, intervals should be as narrow as possible while still satisfying EOC.

We consider the regression task as it is a more general fair ML setting [13, 14] with minimal assumptions about the underlying data distribution.

Achieving EOC confronts various challenges. Firstly, the majority of prior fair ML approaches are developed for classification problems [12, 15], with only a few for regression [14, 16]. This necessitates developing effective techniques to measure and improve EOC in a regression setting. Secondly, ensuring EOC and marginal coverage rate for test data is difficult when true labels are unknown. Finally, prior works [11, 17, 18, 19] have primarily focused on improving fairness and satisfying coverage rate guarantees, but often neglect the width of prediction intervals during the optimization process. This often results in generating wider intervals that limit the amount of decision-making information available. However, optimizing interval width with fairness constraints is a non-convex problem, and coverage rates are difficult to guarantee on noisy data, making it challenging to consider all these factors simultaneously.

To address these challenges, we propose Binned Fair Quantile Regression (BFQR), a distribution-free post-processing method that improves EOC while maintaining a desired marginal coverage rate and a narrow prediction interval. It consists of three major steps: First, a hold-out (calibration) dataset is calibrated to improve EOC based on true label $Y$ within discretized bins; second, we leverage conformal prediction [20, 21] to achieve EOC for test data from the calibration results in the first step; and finally, an efficient and robust optimization technique is developed to minimize the mean width of prediction intervals. Experiments on both synthetic and real-world data show that BFQR is more effective in improving EOC fairness than state-of-the-art methods.

**Related Work.** In regression settings, equal opportunity [12] has been studied mostly by adversarial training [22, 23]. For quantile regression, models with equal opportunity constraint [24, 25] are proposed. However, all mentioned are in-processing methods and have trade-offs between accuracy and fairness [26, 27]. There are several works on uncertainty-aware fairness in regression. The pioneering work equalized coverage [11], is built on the validity of conditional coverage [28], and has several follow-ups [17, 18]. [19] also proposes conformalized fair regression, imposing demographic parity fairness on prediction interval bounds. However, neither of these methods considers the fine-grained group (e.g., low-income females) fairness conditional on true labels as well as the increased width of prediction intervals that provide little information.

## 2 Preliminary

**Conformal Prediction** extends traditional ML by providing a set of prediction intervals around the predicted value, which can be used to assess the level of confidence or uncertainty in the prediction [20]. It is distribution-free and has a rigorous statistical guarantee. A commonly used approach is split conformal prediction [20, 21]. To start, the training data is divided into two sets: the training set $\mathbb{D}_{tr}$ and the calibration set $\mathbb{D}_c = \{(X_1, Y_1), ..., (X_n, Y_n)\}$. A prediction model $\hat{f}$ is trained on the training set $\mathbb{D}_{tr}$. The key ingredient is the conformity score function $S(x, y) \in \mathbb{R}$ used to evaluate the model's prediction performance. Given a desired error rate $\alpha$, it then calculates the quantile $\hat{Q}_{1-\alpha}(S, \mathbb{D}_c)$, denoting the $(1 - \alpha)(1 + 1/|\mathbb{D}_c|)$-th quantile of the empirical distribution of $S$ on $\mathbb{D}_c$. Finally, for a sample $X_{n+1}$ in the test set $\mathbb{D}_t$, its prediction set is $C^S(X_{n+1}) = \{y : S(X_{n+1}, y) \leq \hat{Q}_{1-\alpha}(S, \mathbb{D}_c)\}$. This set contains all possible values of $y$ for which the conformity score $S(X_{n+1}, y)$ is less than or equal to the calculated quantile. Given a mild assumption that the test and calibration set are exchangeable, the coverage rate in conformal prediction is guaranteed with a probability of $P(Y_{n+1} \in C^S(X_{n+1})) \geq 1 - \alpha$.

**Equalized Coverage.** Let $V \in \mathcal{V}$ be the indicator of whether $Y$ is covered in the prediction set $\hat{C}(X)$, i.e., $V = \mathbb{1}[Y \in \hat{C}(X)]$. $\mathcal{V} = \{0, 1\}$. Given a desired error rate $\alpha \in [0, 1]$, equalized coverage is satisfied [11] when $\forall a \in A, Pr\{V|A = a\} = Pr\{V\} \geq 1 - \alpha$ and $Pr\{V\} \geq 1 - \alpha$. Equalized coverage guarantees equal conditional coverage, i.e., $V$ is independent of $A$ (denoted as $V \perp\!\!\!\perp A$), but fails to ensure equal coverage conditional on $Y$, especially for extreme values of $Y$ (Fig. 1). This is problematic as it can perpetuate discrimination against marginalized groups (e.g., females with low income) in risk assessment.

## 3 Equal Opportunity of Coverage

Equal Opportunity of Coverage addresses the limitation of equalized coverage and is defined based on the equal opportunity [12]:

**Definition 3.1** (Equal Opportunity of Coverage (EOC)). EOC is satisfied when $Pr\{V|A = a, Y = y\} = Pr\{V|Y = y\}$, i.e., $V \perp\!\!\!\perp A|Y$ and $Pr\{V\} \geq 1 - \alpha, \forall y \in Y$ and $\forall a \in A$.

The definition of EOC requires that (1) conditioned on the target variable $Y$, whether a sample is covered in its prediction interval should be independent of its sensitive attribute $A$; (2) the marginal coverage rate is above the desired level. Interestingly, the difference between equalized coverage and EOC in their mathematical forms share similarities with the difference between demographic parity and equalized odds. Note that our focus here is whether the true label is covered in the prediction interval since even though the prediction interval contains false labels, $V = \mathbb{I}[Y \in \hat{C}(X)] = 1$ is still valid. Therefore, while EOC has a similar formation to equalized odds, it in fact describes equal opportunity which focuses on $Y = 1$.

Preferably, both EOC and equalized coverage should be guaranteed. However, the *mutual exclusivity theorem* below suggests that there is an inherent trade-off between EOC and equalized coverage:

**Theorem 3.1** (Mutual Exclusivity). *If $A \not\perp\!\!\!\perp Y$ and $V \not\perp\!\!\!\perp Y$, then either equalized coverage or equal opportunity of coverage holds but not both.*

*Proof.* If $V \perp\!\!\!\perp A$ and $V \perp\!\!\!\perp A|Y$, then either $A \perp\!\!\!\perp Y$ or $V \perp\!\!\!\perp Y$.

Unfairness often arises from the fact that features predictive of $Y$ are also correlated to the protected attribute due to e.g., historical bias in the data [29]. This indicates $V \perp\!\!\!\perp A$. For the second condition, though $V \not\perp\!\!\!\perp Y$ is possible, $V$ needs to depend on $Y$ to ensure prediction intervals with reasonable width. Predicting certain values (e.g., extreme values) of $Y$ can be challenging due to representative bias. If we enforce the predictor to provide high coverage for marginalized groups with these values, it is highly likely to result in extremely wide intervals that offer little guidance in decision-making.

**Meauring EOC**. Given the underlying distribution $p \in \Delta(\mathcal{V} \times \mathcal{A} \times \mathcal{Y})$, where $\mathcal{V}, \mathcal{A}, \mathcal{Y}$ is the domain of $v, a, y$ respectively. We can determine how likely $p$ satisfies EOC by measuring its distance to $p'$, the closest distribution that perfectly achieves EOC. Formally, we denote $P_{EOC}$ as the property of EOC, the set of all distributions in defined space that satisfy EOC, i.e., $P_{EOC} := \{p \in \Delta(\mathcal{V} \times \mathcal{A} \times \mathcal{Y}) : (V, A, Y) \sim p, V \perp\!\!\!\perp A|Y\}$. $p' \in P_{EOC}$ is the distribution with minimum total variation (TV) distance to $p$, i.e., $\forall q \in P_{EOC}, d_{TV}(p, p') \leq d_{TV}(p, q)$. The TV distance between $p$ and $p'$ is

formally defined as $d_{TV}(p, p') = \frac{1}{2} \sum_{(v,a,y) \in \Delta(\mathcal{V} \times \mathcal{A} \times \mathcal{Y})} |p(v,a,y) - p'(v,a,y)| = \frac{1}{2} \|p - p'\|_1$, where $\|\cdot\|_1$ denotes the $l_1$ norm of a distribution. Instead of the commonly-used Kolmogorov–Smirnov (KS) distance [12, 30], TV distance is chosen as the measure due to the significant drawback of the KS distance – its insensitivity to the deviations between $p$ and $p'$ at the tails [31]. Whereas in real-world data, we often confront such significant deviation (see e.g., Fig. 1).

It is challenging to directly measure the distance between $p$ and $p'$ from observed data when the target variable $Y$ is continuous due to data sparsity issues for each value of $Y$. However, drawing from previous works [32, 33], we can instead use an easy-to-compute statistic $T$ as a surrogate of $d_{TV}(p, p')$. $T$ is the measure of independence, assessing the weighted summed violation of independence within each discretized bin. The expectation of $T$, $\mathbb{E}[T]$, has a fixed upper bound when $p$ satisfies EOC, and a lower bound that increases with the TV distance between $p$ and $p'$. The intuition here is that when $Y$ is divided into sufficient bins and $p_Y := Pr\{V, A|Y\}$, the conditional distribution of $p$ on $Y$, is Lipschitz continuous, enforcing independence within each bin generates a distribution that does not deviate far from $P_{EOC}$ in terms of TV distance.

**Lemma 3.2.** *[33] Let $d = \lceil n^{2/5} \rceil$ be the number of bins $Y$ is discretized into, with each bin having an equal size of samples. Let $T$ be a sum of independence measures within all discretized bins. Given the Lipschitz continuity of $p_Y$ in Assumption 2, $L$ is the Lipschitz constant, we have:*

*1) When $p$ satisfies EOC, $\mathbb{E}[T] \leq \frac{|\mathbb{D}|L^2}{d^2}$;*

*2) When $d_{TV}(p, p') = \epsilon$, there exists a constant $Z$ such that $\mathbb{E}[T] \geq Z(\epsilon - 3\frac{L}{d})^2$.*

Lemma 3.2 indicates that we can approximately evaluate EOC for $p$ by measuring the independence within each bin. If $V \perp\!\!\!\perp A$ almost holds in each bin, such that $\mathbb{E}[T]$ is small enough, then it is highly possible that $V \perp\!\!\!\perp A|Y$, i.e., $p$ satisfies EOC. Moreover, an increasing function of $\epsilon$ is upper bounded by $\mathbb{E}[T]$. As $\mathbb{E}[T]$ increases, it is highly likely that $p$ is further away from any $p'$ that satisfies EOC. This lemma serves as a theoretical foundation for the proposed method below to improve EOC and for adopting $T$ as an evaluation metric in experiments.

In order to calculate $\mathbb{E}[T]$ from data, we introduce unbiased estimators of $T$ in Appendix 8.1.1. According to the central limit theorem, we could estimate $\mathbb{E}[T]$ through a sufficient number of random samplings. Whereas for the sake of efficiency, we prefer to construct $T$ with bounded variance such that $\mathbb{E}[T]$ could converge through limited repeated samplings.

## 4 Improving Equal Opportunity of Coverage

In this section, we introduce a post-processing approach where we have a trained ML model, a calibration dataset, and a test dataset for which we aim to improve EOC. It consists of three steps. First, we enforce the independence of $A$ and $V$ within each interval of $Y$ (i.e., EOC) for the calibration data, and then leverage conformal prediction to achieve EOC for test data. Lastly, we describe an efficient and robust optimization approach that optimizes both EOC and widths of prediction intervals.

### 4.1 Improving EOC on calibration data

The first step aims to improve EOC on the calibration data where we have ground-truth labels. As the target variable $Y$ is continuous, the number of samples with certain values of $Y$ can be extremely small, thus calibrating for each distinct value of $Y$ is almost impossible. Meanwhile, commonly used methods designed for continuous variables such as adversarial learning [23, 22], which intend to learn a near-optimal $p$, are computationally inefficient for post-processing approaches [34].

According to Lemma 3.2, $\mathbb{E}[T]$ is an upper bound of an increasing function of $d_{TV}(p, p')$. Thus, this indicates that if $\mathbb{E}[T]$ decreases, the maximum distance between $p$ and $p'$ is reduced, resulting in an improvement in EOC. Informed by this idea, we first focus on enhancing EOC on the calibration data $\mathbb{D}_c$. For simplicity, we employ the framework introduced in [35] as the conformal prediction model, though our post-processing method is applicable to any base model. Specifically, let $\hat{q}_\alpha$ denotes the $\alpha$-th conditional quantile regression function, i.e., for $i$-th sample $(X_i, Y_i)$, $\hat{q}_\alpha(X_i) := \inf\{y \in \Delta\mathcal{Y} : Pr\{Y_i \leq y|X = X_i\} \geq \alpha\}$. Fix the lower and upper quantiles as $\alpha_{lo} = \alpha/2$ and $\alpha_{hi} = 1 - \alpha/2$, then $\hat{q}_{\alpha_{lo}}(X_i)$ and $\hat{q}_{\alpha_{hi}}(X_i)$ denote lower and upper quantile regression functions, respectively. The base model is trained on the training data and used for inference on calibration and test data.

Following the discretization idea in Lemma 3.2, we divide the continuous variable $Y \in [Y_{min}, Y_{max}]$ in $\mathbb{D}_c$ into $M$ bins with equal sample sizes, and the $m$-th bin is denoted as $B_m = [Y_m^-, Y_m^+)$.

To enhance EOC on $\mathbb{D}_c$, we can minimize $\mathbb{E}[T]$ by enforcing the independence of $A$ and $V$ within all discretized bins. In particular, we fix the coverage rates within each bin as $\beta_m \in [0, 1]$, therefore $V \perp\!\!\!\perp A | Y \in B_m$. Note that, although the coverage rates are equal across groups within the same bin, the quantile value at $\beta_m$ are computed separately for each group, as illustrated in Figure 2. We first calculate the vanilla prediction intervals $\hat{C}(X_i) = [\hat{q}_{\alpha_{lo}}(X_i), \hat{q}_{\alpha_{hi}}(X_i)]$ obtained from the trained model, and get the conformity score $S(X_i, Y_i) = \max(\hat{q}_{\alpha_{lo}}(X_i) - Y_i, Y_i - \hat{q}_{\alpha_{hi}}(X_i))$ for all samples in $\mathbb{D}_c$. Then for different combinations of bins and protect groups, we calculate the quantile value of conformity scores at coverage rate $\beta_m$, $\hat{Q}_\beta(S, \mathbb{D}_c(a, m))$, for data in $\mathbb{D}_c(a, m) = \{i | i \in \mathbb{D}_c, A_i = a, Y_i \in B_m\}$. To simplify notations, we substitute $\hat{Q}_\beta(S, \mathbb{D}_c(a, m))$ by $G_{a,m}(\beta_m)$. Since each bin has an equal sample size and coverage rate $\beta_m$, the average coverage rate $\sum_m \beta_m / M$ is then set to $1 - \alpha$ to keep a coverage rate of $1 - \alpha$ on the calibration data.

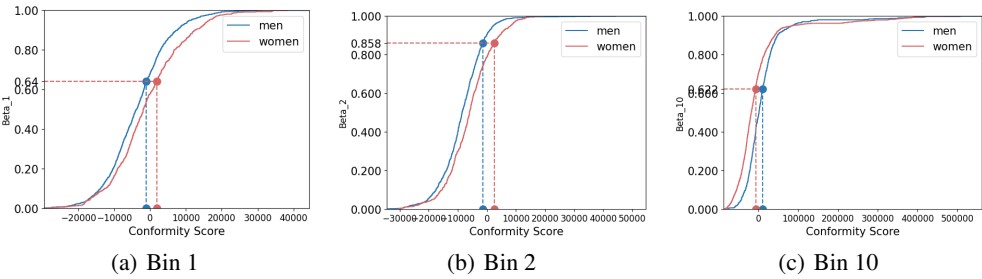

(a) Bin 1        (b) Bin 2        (c) Bin 10

Figure 2: Empirical Cumulative Distribution Functions (ECDF) of conformity scores of men (blue) and women (red) within bins 1,2 and 10 on the *Adult* dataset. The desired coverage rates within different bins could be set at different levels but should be equal across groups within the same bin, as indicated by the overlapping horizontal dashed lines. Due to the distinct disparity between the ECDF of men and women, the same coverage rate is mapped to different quantile values on the x-axis.

Through the reconstruction of prediction intervals $C(X_i)$ with $G_{a,m}(\beta_m)$, i.e., $\forall i \in \mathbb{D}_c(a, m)$, $C(X_i) = [\hat{q}_{\alpha_{lo}}(X_i) - G_{a,m}(\beta_m), \hat{q}_{\alpha_{hi}}(X_i) + G_{a,m}(\beta_m)]$, EOC is enhanced and the marginal coverage rate is guaranteed on the calibration data.

## 4.2 For Coverage and Independence Guarantees

This subsection seeks to preserve EOC and marginal coverage rate for test data based on results for calibration data through conformal prediction. The key is to find out, for a test sample with $A_{n+1} = a_{n+1}$, which bin $B_m$ it belongs to. We can then calibrate the test sample with the quantile $G_{a,m}(\beta)$. However, direct calibration based on the predicted bin would not improve EOC since the prediction result can be biased due to, e.g., skewed distributions for different groups [29].

To address this issue, we propose Binned Fair Quantile Regression (BFQR) (see Algorithm 1 in Appendix 8.2.1). Our method could be treated as a variant of Mondrian conformal prediction [36, 37], where the confidence in each bin is evaluated independently. Suppose that a new data point with feature $X_i$ and protected attribute $a$ falls into a certain bin $B_m$, we calibrate it with the corresponding quantile value $G_{a,m}(\beta_m)$. Then, we obtain a sub-interval of prediction within bin $B_m$, i.e., $C_m(X_i) = B_m \cap [\hat{q}_{\alpha_{lo}}(X_i) - G_{a,m}(\beta_m), \hat{q}_{\alpha_{hi}}(X_i) + G_{a,m}(\beta_m)]$. After computing $C_m(X_i)$, a union of all subsets $C(X_i) = \bigcup_m C_m(X_i)$ is then the prediction interval of $X_i$. Under a mild exchangeability assumption similar to [11], our algorithm provides both the marginal coverage guarantee and fair coverage guarantees within each bin.

*Assumption* 1 (Exchangeability). All calibration data $(X_i, Y_i)$, $i = 1, ..., n$ and a sample of test data $(X_{n+1}, Y_{n+1})$ are exchangeable conditioned on $A_{n+1} = a$ and $Y_{n+1} \in B_m$, and conformity scores $\{S(X_i, Y_i), i \in \mathbb{D}_c(a, m) \cup \{n + 1\}\}$ are almost surely distinct.

**Theorem 4.1** (Bin Coverage and Independence Guarantee). *Under Assumption 1, $\beta_m \leq Pr\{Y_{n+1} \in C(X_{n+1}) | A_{n+1} = a, Y_{n+1} \in B_m\} \leq \beta_m + 1/(|\mathbb{D}_c(a, m)| + 1)$. The expectation of max coverage gap inside $m$-th bin is upper bounded by $\max_a\{1/(|\mathbb{D}_c(a, m)| + 1)\}$.*

**Theorem 4.2** (Marginal Coverage Guarantee). *If we have $\sum_m \beta_m / M = 1 - \alpha$, under Assumption 1, then $Pr\{Y_{n+1} \in C(X_{n+1})\} \geq 1 - \alpha$.*

Here we sketch the proofs. Take $(X_i, A_i, Y_i)$ as new sample from test data, $i \in \mathbb{D}_t$. Suppose $Y_i \in B_m$, then the $p$-value of the null hypothesis $Y_i \in B_m$ is given by $\hat{u}_{A_i,m} = \frac{1+|j \in \mathbb{D}_c(A_i,m):S(X_j,Y_j) \leq S(X_i,Y_i)|}{1+|\mathbb{D}_c(A_i,m)|}$, which is the proportion of conformity scores that are less than the score of the new sample among all calibration data that have the same protected attribute and fall into the same bin. The prediction interval $\hat{C}_m(X_i)$ in $B_m$ is the intersection of $B_m$ and the prediction interval calibrated by $G_{a,m}(\beta_m)$, which includes the part of $B_m$ in the prediction interval where the $p$-value $\hat{u}_{A_i,m}$ is greater than $\beta_m$. This guarantees the bin coverage of $B_m$ at level $\beta_m$ in Theorem 4.1. The complete proofs can be found in Appendix 8.2.2.

### 4.3 Constrained Optimization

With improved EOC and coverage guarantee, we now need to identify $\beta_m$ for each bin so that the mean width of prediction intervals for $\mathbb{D}_t$ is the smallest. As such, our goal is to solve a constrained optimization problem, where the decision variables are the coverage rates in each bin $\beta_m$, $m = 1, \ldots, M$, and the objective function is the mean width of prediction intervals in test data:

$$\min_{m,i \in |\mathbb{D}_t|} \sum |C_m(X_i)|/|\mathbb{D}_t| \qquad (1)$$
$$\text{s.t.} \quad \begin{cases} \sum_m \beta_m/M = 1 - \alpha, \\ \beta_m \in [0,1], \quad m = 1, \ldots, M. \end{cases}$$

An easy solution to the optimization problem Eq. 1 can be obtained by adjusting $\hat{C}(X_i)$, $X_i \in \mathbb{D}_c$ with split conformal prediction described in Section 2. However, it is not optimal since $\beta_m$ is determined by $\hat{Q}_{1-\alpha}(S, \mathbb{D}_c)$ calculated on all calibration data, but different bins have varying costs of width associated with changes in their coverage rates. For instance, if a bin has a coverage rate of 0.98, increasing it to 0.99 would lead to a significant increase in its width, whereas increasing the coverage rate of a bin from 0.50 to 0.51 would result in only a minor increase in width. Therefore, we use the solution of split conformal prediction as the initialization and then optimize it.

Solving Eq. 1 is challenging in that it is computationally expensive or even infeasible to compute the value and the gradient of the objective function. First, the computation of prediction interval $C(X_i)$ involves multiple intersection and union operations, which is a complex step function of $\beta_m$. Second, prediction interval $C(X_i)$ is related to quantile value $G_{a,m}$, which is estimated from data with noise. Directly using slopes of $G_{a,m}$ as the gradient methods could result in over-fitting to noise, as shown in Section 5.2.4. Third, We cannot assume the objective function's convexity or differentiability as the data may come from any possible distribution, and sorting is involved in calculating $G_{a,m}$ [38]. To address those obstacles, we propose an efficient and robust optimization algorithm (detailed in Algorithm 2 in Appendix 8.3.2) that utilizes a relaxed upper bound and optimizes through approximated subgradients [39].

The steps of our approach are described below. First, to accelerate computing, we use a dummy continuous prediction interval $C^d(X_i) = Convex(\bigcup_m C_m(X_i))$, i.e., the convex hull of all sub-intervals, as an upper bound to substitute the original prediction interval $C(X_i)$ in the objective function. However, $C^d(X_i)$ is still related to noisy $G_{a,m}$. To address this, in each iteration, we compute the slope for each bin $m$ in increasing and decreasing directions, denoted as $\hat{t}_m^+$ and $\hat{t}_m^-$, respectively. Since the objective function aims to decrease without changing the marginal coverage rate, we take a greedy strategy by moving up a step $\eta$ in the bin with the steepest descendent direction $\max_m\{\hat{t}_m^-\}$, meanwhile taking a step $\eta$ down the slowest ascendant direction $\min_m\{\hat{t}_m^+\}$. We stop until $\max_m\{\hat{t}_m^+\} \geq \min_m\{\hat{t}_m^-\} + 2\varepsilon$, where $\varepsilon$ is an appropriate estimation error bound related with $|\mathbb{D}_c|$ [40, 41]. More details can be found in Appendix 8.3.1.

The proposed method can be viewed as a subgradient method, incorporating considerations for quantile value estimation errors and maintaining a constant mean coverage rate. Combining the three steps, we could get prediction intervals with improved EOC, guaranteed marginal coverage, and decreased average width of prediction intervals.

## 5 Experiments

In this section, we conduct three sets of experiments on both synthetic and real-world data to evaluate (1) the effectiveness of the proposed approach for achieving EOC (Section 5.2) ; (2) the impact of the

key parameter $M$, the number of bins (Section 5.2.3); and (3) the effectiveness and efficiency of the proposed optimization framework (Section 5.2.4).

## 5.1 Experimental Setup

We propose two metrics to evaluate the first property of EOC, i.e., whether coverage rates for different groups with similar outcomes are close. With the $M$ discretized bins in Section 4.1, the first metric is the average of maximum difference in coverage rates between groups for all bins, similar to the definition of conditional KS distance in [12]. In main experiments, $M$ is set as 20. The second metric $T$ is introduced in Lemma 3.2, where $T$ is calculated on $d = \lceil |\mathbb{D}_t|^{2/5} \rceil$ bins. Specifically, $d \neq M$ for fairness of evaluation, e.g., $d = 100$ for synthetic data. A smaller $T$ implies a closer distance to the ideal EOC distribution and, therefore a better EOC. Furthermore, for the second property of EOC, we need to ensure the desired marginal coverage rates. The efficiency of uncertainty estimation is measured by the width of the prediction interval. In addition, we check the conditional coverage rates on groups to measure equalized coverage. All metrics related to coverage rates are multiplied by 100 in tables to exhibit significant differences. Every experiment is repeated 100 times on random divisions of data with different seeds, with $|\mathbb{D}_{tr}| : |\mathbb{D}_c| : |\mathbb{D}_t| = 3 : 1 : 1$.

We compare our method with the following state-of-the-art methods: 1) Split Conformalized Quantile Regression (CQR) [35, 21] with only marginal coverage guarantee; 2) Group-conditional Conformalized Quantile Regression (GCQR) [11] with both marginal and conditional coverage guarantee; 3) MultiValid Predictor (MVP) [18] with both marginal and conditional coverage guarantee. Note that the base model for this algorithm is trained on the union of training and calibration data as MVP does not require any calibration data; 4) Conformal Fair Quantile Prediction (CFQR) [19], which guarantees marginal coverage and demographic parity on both upper and lower bounds of the predicted intervals; 5) Label-conditional Conformalized Quantile Regression (LCQR) [42], which is designed for classification problems to provide marginal coverage and equalized coverage for each class. We adapt it to regression tasks. The base model for all compared conformal prediction methods is set as the QR model at the level of 0.05 and 0.95, and the desired marginal coverage is set to 0.9. Considering that for some real-world applications like scoring [43], disjoint prediction intervals make little sense, we evaluate prediction intervals $C(X)$ along with their dummy prediction intervals $C^d(X)$ in Section 4.3, represented as BFQR and BFQR*. In the comparison tables, the best results and the second-best results are highlighted in bold and underlined, and undercovered groups who fail to reject the null hypothesis at 0.05 level in one-tailed t-tests are emphasized in Italian.

## 5.2 Results

### 5.2.1 Synthetic data

We generate ten independent and exponentially distributed features with the scale of 1, $X = (X_1, \ldots, X_{10})$; protected attribute $A$ is randomly selected from $\{0, 1, 2\}$ with a probability of 0.1, 0.2, 0.7, respectively. The labels $Y$ for $A = 1$ follow a random distribution, thus impossible to predict; labels $Y$ for the other two groups are the linear summations of $X$ and $A$, plus noises that increase with $Y$. A size of 100,000 samples are generated from this distribution and the data generating process is detailed in Appendix 8.4.1.

We have the following observations from the results in Table 1: 1) All methods have marginal coverage guarantee, which is attributed to statistical guarantee from conformal prediction. 2) Our proposed methods, BFQR with disjoint intervals and BFQR* with joint intervals achieve the best trade-off between EOC and equalized coverage. In particular, our method achieves the second-best EOC (i.e., Mean Max Coverage Gap and $T$), meanwhile, the conditional coverage rates are almost equal across different groups. While CFQR has significantly better performance w.r.t. EOC, the conditional coverage for $A = 1$ (the most challenging case) is extremely low compared to $A = 0$ and $A = 2$. This result aligns with the mutual exclusivity between EOC and equalized coverage formulated in Theorem 3.1. 3) Among all methods, the average interval width of our method is the smallest, validating the effectiveness of the optimization process in Section 4.3. One of the main advantages of BFQR and BFQR* is optimizing through bin coverages. In this process, bins that sacrifice interval width for an over-coverage rate are adjusted to a lower but satisfactory coverage rate. Therefore, we are able to guarantee a smaller average interval width while improving the EOC.

Table 1: Experiments results for synthetic data.

| Method | EOC | | | Width ↓ | Equalized Coverage | | |
|---|---|---|---|---|---|---|---|
| | Mean Max Coverage Gap ↓ | $T$ ↓ | Marginal Coverage | | Coverage $(A=0)$ | Coverage $(A=1)$ | Coverage $(A=2)$ |
| CQR | 20.11±1.05 | 306.33±18.78 | 89.99±0.32 | 16.02±0.04 | *82.84±0.90* | 90.03±0.76 | 91.00±0.29 |
| GCQR | 14.48±1.36 | 386.81±35.60 | 90.00±0.33 | 16.21±0.06 | 90.08±0.99 | 89.92±0.64 | 90.01±0.37 |
| MVP | 9.03±1.43 | 71.27±6.70 | 89.98±0.32 | 16.31±0.13 | 89.35±0.94 | 90.11±0.54 | 89.96±0.24 |
| CFQR | **0.19±0.09** | **0.09±0.24** | 89.93±0.20 | 17.00±0.06 | 93.89±0.53 | *71.38±0.87* | 94.74±0.24 |
| LCQR | 6.38±0.26 | 18.64±6.71 | 90.41±0.29 | 17.32±0.09 | 91.37±0.84 | 90.55±0.68 | 90.23±0.37 |
| BFQR* | 3.15±0.58 | 2.78±3.79 | 91.74±1.28 | 16.24±0.21 | 93.81±1.03 | 91.19±3.13 | 91.60±0.90 |
| BFQR | 3.82±0.45 | 3.65±4.75 | 90.03±0.32 | **15.96±0.13** | 91.99±1.16 | 89.08±3.61 | 90.03±1.04 |

### 5.2.2 Real-world Data

We further evaluate our method on two benchmark datasets: *Adult* [44, 45] where gender is the protected attribute and the outcome is salary; *MEPS* (Medical Expenditure Panel Survey) data [46, 11] where race is the protected attribute and the outcome is the health care system utilization score.

We observe similar results: For Adult data (shown in Table 2), all methods achieve marginal coverage. Our methods achieve the best EOC and smallest mean width of prediction interval while maintaining competitive conditional coverage rates. CFQR with the best EOC on synthetic data does not have consistently good performance, and the mean interval width is greatly larger compared with other methods. The increased width of LCQR implies that as some bins are difficult to predict, enforcing all bins to reach the same high coverage rates generates prediction results with little useful information. For MEPS data (shown in Table 3), the results are similar to those in synthetic data: our method is the second-best w.r.t. EOC, with marginal coverage rates guaranteed and a significantly smaller average prediction width.

Table 2: Experiments results for *Adult* data.

| Method | EOC | | | Width ↓ | Equalized Coverage | |
|---|---|---|---|---|---|---|
| | Mean Max Coverage Gap ↓ | $T$ ↓ | Marginal Coverage | | Coverage (Men) | Coverage (Women) |
| CQR | 5.01±0.40 | 67.31±10.56 | 89.98±0.29 | 93,951.11±409.25 | 90.01±0.42 | 89.94±0.37 |
| GCQR | 4.97±0.50 | 67.30±10.63 | 89.99±0.29 | 93,994.10±435.35 | 89.98±0.41 | 89.99±0.46 |
| MVP | 5.50±0.34 | 211.01±19.44 | 90.05±0.28 | 98,229.81±1,723.82 | 90.07±0.13 | 90.08±0.27 |
| CFQR | 3.57±0.53 | 18.17±6.02 | 90.08±0.14 | 147,253.27±499.47 | 90.02±0.42 | 90.08±0.40 |
| LCQR | 3.91±0.45 | 5.03±4.42 | 90.50±0.30 | 160,107.96±8929.07 | 90.46±0.37 | 90.54±0.46 |
| BFQR* | **2.90±0.37** | **3.60±3.29** | 91.08±0.46 | 93,689.33±976.76 | 91.89±0.74 | 90.11±0.43 |
| BFQR | 3.05±0.35 | 3.55±3.14 | 90.32±0.28 | **91,969.66±996.63** | 90.97±0.42 | 89.56±0.47 |

Table 3: Experiments results for *MEPS* data.

| Method | EOC | | | Width ↓ | Equalized Coverage | |
|---|---|---|---|---|---|---|
| | Mean Max Coverage Gap ↓ | $T$ ↓ | Marginal Coverage | | Coverage (White) | Coverage (Non-white) |
| CQR | 6.47±1.22 | 7.62±4.28 | 89.85±0.77 | 32.06±1.85 | 89.81±0.99 | 89.90±0.88 |
| GCQR | 6.54±1.79 | 9.50±8.35 | 89.91±0.74 | 32.08±1.90 | 89.91±0.95 | 89.93±1.11 |
| MVP | 8.27±2.37 | 8.78±3.47 | 89.95±0.84 | 41.08±6.87 | 90.97±0.71 | 90.75±0.97 |
| CFQR | **0.94±0.39** | **0.28±0.55** | 90.88±0.51 | 36.55±1.85 | *87.93±1.06* | 93.18±0.78 |
| LCQR | 5.29±0.97 | 3.29±2.52 | 91.97±0.69 | 160.65±30.71 | 91.59±0.94 | 92.57±1.02 |
| BFQR* | 3.04±0.64 | 1.20±1.70 | 92.33±0.75 | 23.95±2.53 | 93.83±0.84 | 89.94±1.16 |
| BFQR | 3.99±0.76 | 2.27±2.08 | 91.05±0.80 | **23.11±2.41** | 92.66±0.85 | *88.46±1.34* |

### 5.2.3 The Impact of $M$

The number of bins, $M$, is the only primary parameter in our method. Using synthetic data that ideally satisfies Assumption 1, we evaluate how $M$ influences EOC. In our experiment, $M$ varies

among {1, 5, 10, 20, 50}. When bin size is 1, our method degenerates to GCQR. The results are shown in Fig. 3, where the best EOC for BFQR and BFQR* is achieved with 10 bins. When the number of bins is small, the sample size of $\mathbb{D}_{a,m}$ increases. According to Theorem 4.1, the upper bound of the expectation of max group coverage rate gap is decreased. However, this only suggests a decrease in the expectation, not for every $Y$ within bins. When the quantile value is calculated on large samples, it fails to characterize the conformity scores of individuals. BFQR* with continuous intervals exhibits better EOC compared to BFQR as its coverage gaps are primarily influenced by the quantile values of the first and last bins, involving less randomness. As $M$ greatly affects the performance of our methods, it should be cautiously chosen for various problem settings.

### 5.2.4 Comparisons of Optimization Methods

To demonstrate the efficiency of the proposed optimization method, we compare it with three recent optimization methods: 1) BFGS-SQP [47, 48], a constrained non-smooth optimization method, 2) Augmented Lagrange (AL) [49], another method to solve constrained non-smooth optimization problems, and 3) Bayesian optimization [50, 51], a global optimization method for solving problems with noisy objective functions. We evaluate these methods based on their performance in terms of EOC, width, and running times. The results on synthetic data are in Table 4. The Bayesian method has extremely slow computational speed and cannot guarantee marginal coverage. In comparison to BFGS-SQP and AL, our method achieves similar EOC performance but with slightly lower prediction interval widths, which confirms the effectiveness of our subgradient approximation. Additionally, our method is more efficient as it significantly reduces running times.

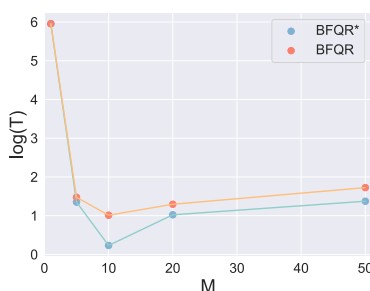

Figure 3: Impact of $M$ on EOC.

Table 4: Optimization results on synthetic data.

| Optimzation Method | EOC | | | Width ↓ | Running Time (seconds)↓ |
|---|---|---|---|---|---|
| | Mean Max Coverage Gap ↓ | $T$ ↓ | Marginal Coverage | | |
| BFQS-SQP | 4.86±0.75 | 3.89±3.71 | 90.08±0.16 | 16.15±0.06 | 78.11±41.98 |
| AL | 4.81±0.73 | 3.83±3.67 | 90.18±0.18 | 16.16±0.06 | 77.57±1.50 |
| Bayesian | 8.41±1.96 | 4.47±5.63 | *88.18±10.08* | 18.80±2.51 | 805.81±152.05 |
| BFQR | **3.82±0.45** | **3.65±4.75** | 90.03±0.32 | **15.96±0.13** | **33.62±14.78** |

## 6 Discussion

**EOC or equalized coverage?** The results in Section 5.2 clearly demonstrate that perfect EOC and equalized coverage are mutually exclusive, empirically verifying Theorem 3.1. In real-world applications, it becomes crucial to trade off between EOC and equalized coverage. We recommend placing a higher emphasis on EOC when certain labels are considered more favorable, e.g., when individuals labeled as low-salary are the most important subpopulation for decision-making. Under EOC, each group with the same label is treated equally, not only in terms of equal coverage rates (i.e., equal probability of being included in the prediction result) but also in terms of comparable prediction interval widths as a measure of uncertainty. For example, under equalized coverage, the coverage rate for men is around 0.65 but 0.55 for women in the first bin, low-income population. We cannot conclude that the model is more confident in the prediction results for low-income men solely based on the larger prediction intervals of men: this discrepancy might arise from the over-coverage of men. In this sense, our designed metric EOC, not only contributes to evaluating and enhancing fairness under uncertainty but also serves for fair uncertainty quantification, discovering model bias in uncertainty quantification. A possible application of EOC is for guiding sample selection in active learning [52, 53], allowing the applied model to label the most uncertain samples in a fair manner.

**Discretizition** into equal-mass bins is a favorable strategy to avoid excessively small sizes of $\mathbb{D}(a,m)$. While this guarantees a tighter upper bound for the expectation of the maximum coverage gap within bins according to Theorem 4.1, our method can be extended beyond equal-mass bins. When the

number of samples in each bin is unequal, we adjust the constraints in Eq. 1 by incorporating a weighted sum of $\beta_m$, i.e., $\sum_{m \in [M]} \frac{\beta_m}{M|\mathbb{D}(m)|} = 1 - \alpha$. Consequently, in Algorithm 2, the gradient in the direction of each bin also should be weighted by $|\mathbb{D}(m)|$. These adaptations allow our method to maintain its effectiveness when the sample sizes in each class are uneven, e.g., in imbalanced classification problems. Moreover, our method is able to address certain coverage rate concerns within particular bins, such as low-income females, by assigning specific coverage rates to the corresponding bins while allowing flexibility in adjusting the coverage rates of other bins.

**Why Better EOC?** Our methods have been empirically shown to have better EOC than methods designed for equalized coverage, and here we elucidate the reasons behind this improvement. Given the underlying distribution of the $\alpha$-quantile conformity scores in group $a$ conditioned on $y$ as $Q_{a,y}(\alpha)$. For every $y$ and $a$, calibrating with $Q_{a,y}(\alpha)$ would generate perfect EOC. However, due to the limitations of available calibration samples, the estimation of $Q_{a,y}(\alpha)$ is subject to bias. Since the base model cannot perfectly capture the underlying distribution, the conformity score, as a measure of disagreement, must correlate with $Y$. While equalized coverage methods generally ignore this correlation, our method tries to handle it. In Section 4.1, we used $G_{a,m}(\beta_m)$ to estimate $Q_{a,y}(\beta_m)$ for all $y \in B_m$, which is the $\beta_m$-quantile value of conformity scores whose $y$ belong to bin $B_m$. The localized estimation within discretized bins is upper-bounded by the largest $\beta_m$-quantile value of score conditioned on any $y$ and lower-bounded by the smallest one, regardless of the unknown distribution of $Y$. Consequently, we could better characterize the conditional distribution within each bin. This enhancement is especially evident when the Lipschitiz constant $L$ in Lemma 3.2 is small, such that the deviation from perfect EOC to the distribution generated by bins is negligible.

**Possible Extension.** For future work, we aim to further improve our method by seeking a tighter bound to achieve optimal EOC. Possible direction may follow [14, 54], by discretizing $Y$ into finite values and bounding deviation in EOC of the discretized predictor. Additionally, our conformalized method leverages statistical properties to ensure coverage rates in each bin and optimize prediction interval widths, making it particularly appropriate for large datasets. With a better discretization strategy that better utilizes the calibration samples, e.g., ensemble sampling, our method may maintain its effectiveness on a small calibration set.

**Potential Exploration beyond the Current Scope.** The context of this paper is situated in post-processing fairness. Though post-processing methods typically underperform in-processing methods, in-processing methods are not applicable in many situations, e.g., the prediction model is a pre-trained black-box regression model, or a flexible and computationally efficient method is required. Whereas, adapting the proposed method to an in-processing setting to obtain better EOC is a possible extension. It would also be interesting to extend our method to the context that sensitive attributes are not available in the test data, possibly through missing data augmentation for the calibration set [55] or prediction-powered inference [56].

# 7 Conclusion

In this paper, we introduce a new uncertainty-aware fairness notion, equal opportunity of coverage (EOC), which addresses the limitations of the seminal work of equalized coverage [11]. EOC has several desired properties: It guarantees equal coverage rates for groups with the same labels and marginal coverage rate at a pre-determined level. It also ensures a small prediction interval. The theoretical analyses and empirical findings indicate that EOC and equalized coverage are generally incompatible. We suggest using EOC as an alternative to equalized coverage when equal coverage rates and assessment of uncertainty are needed for more fine-grained demographic groups. To improve EOC, we propose a distribution-free post-processing method, BFQR, based on discretization. Experimental results on synthetic and real-world datasets show that BFQR achieves competitive EOC and ensures guaranteed marginal coverage rates with small mean prediction interval widths compared to the state-of-the-art. Moreover, BFQR is adaptable to various settings, such as classification and other decision-making tasks.

## Acknowledgments and Disclosure of Funding

This work is supported in part by NSF under grant III-2106758. Lu Cheng is in part supported by the Cisco Research Gift Grant. We thank Xinhua Zhang for the helpful discussion on the optimization

method. We are grateful to the anonymous reviewers at NeurIPS 2023 for providing valuable feedback and suggestions.

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
