# 8   Supplementary Material

## 8.1   Details and Proofs for the Proposed EOC

### 8.1.1   Calculation of $T$

Given data $\mathbb{D}$, disaggregate $Y$ into $M$ equal-size bins, and the $m$-th bin is denoted as $B_m$. Let $\sigma_m = |B_m|$ denote the number of samples in $B_m$. For distribution $p \in \Delta(\mathcal{V} \times \mathcal{A} \times \mathcal{Y})$ conditioned on $y$ in $B_m$, $p_{V,A|y_m}$, $p_{V|y_m}$ and $p_{A|y_m}$ are denoted as the joint distribution of $(V, A)$, marginal distribution of $V$ and $A$, respectively. Let $U_m$ denote an unbiased estimator of the $L_2^2$ distance between $p_{V|y_m} p_{A|y_m}$ and $p_{V,A|y_m}$, i.e.,

$$\left\| p_{V|y_m} p_{A|y_m} - p_{V,A|y_m} \right\|_2^2 = \sum_{v \in \{0,1\}} \sum_{a \in \Delta \mathcal{A}} (p_{V|y_m}(v) p_{A|y_m}(a) - p_{V,A|y_m}(v,a))^2. \tag{2}$$

As detailed in Section 5.1 of [33] and Algorithm 4 of [32], $U_m$ could be calculated through U-statistic. Specifically, in [33], they consider designing kernel as $\phi_{ij}(av) = \mathbb{I}(A_i = a, V_i = v) - \mathbb{I}(A_i = a)\mathbb{I}(V_i = v)$, for $i$ and $j$-th sample in $\mathbb{D}_t$. Next, they take 4 distinct $i, j, k, l$-th observations from $\mathbb{D}_t$, and calculate the kernel function $h_{ijkl} = \frac{1}{4!} \sum_{\pi \in [4!]} \sum_{a \in \mathcal{A}, v \in \mathcal{V}} \phi_{\pi_1 \pi_2}(av) \phi_{\pi_3 \pi_4}(av)$, where $\pi$ is a permutation of $i, j, k, l$. Afterward, the kernel function is calculated on every 4 distinct samples from $\mathbb{D}_t$. This process is utilized to construct the U-statistic: $U(\mathbb{D}_t) := \frac{1}{\binom{Z}{k}} \sum_{i<j<k<l} h_{ijkl}$, where $Z = |\mathbb{D}_t|$. This U-statistic is an unbiased estimator of $L_2^2$ distance in 2. Poisson sampling is also introduced as an effective technique for reducing computation expenses while maintaining the desired unbiased properties. With this technique, they suggest to calculate $U(\mathbb{D}_t)$ with only $Z_p \sim Poi(Z/2)$ randomly selected samples. However, the algorithm based on the U-statistic still exhibits a time complexity of $O(\sigma_m!)$ in each bin and the expectation of sample number in each bin $\mathbb{E}[\sigma_m] = Z^{\frac{3}{5}}/2$. The total time complexity for each run is at least $O((Z^{\frac{3}{5}}/2)! \cdot Z^{\frac{2}{5}})$, which is too computationally expensive for large datasets.

As an alternative to the U-statistic, we turn to the V-statistic in [57, 58], which could also serve as an unbiased estimator for independence distance. According to the Parseval-Plancherel formula, the Fourier transform of a power of a Euclidean distance, i.e., $L_2^2$ distance, is also a (constant multiple of a) power of the same Euclidean distance. Therefore, we could calculate the energy distance of independence, which is a weighted $L_2^2$ distance between the characteristic function of $V, A|y_m$ and the product of marginal characteristic functions of $V|y_m$ and $A|y_m$. Formally, let $\hat{f}.$ denote the characteristic function of any variable, the energy distance of independence between $V|y_m$ and $A|y_m$ is then defined as $\left\| \hat{f}_{V|y_m} \hat{f}_{A|y_m} - \hat{f}_{V,A|y_m} \right\|_2^2 = \int |\hat{f}_{V|y_m} \hat{f}_{A|y_m} - \hat{f}_{V,A|y_m}|^2 w(a,v) dadv$, where $w(a,v)$ is a weight function. According to [58], this distance is a scaled $L_2^2$ distance multiplied by a constant, which is only related to the dimensions of calculated variables. Since we only utilize distance for comparison and evaluation, the constant could be ignored. Adopting the similar Poisson sampling strategy, the V-statistic of energy distance in a bin of $\sigma_m$ samples could be computed in $O(\sigma_m^2)$. Specifically, in each bin, we first compute two $\sigma_m \times \sigma_m$ distance matrices $M^a$, $M^v$. $M_{i,j}^a$ and $M_{i,j}^v$, the elements $i$-th row and $j$-th column denotes the distance between $i$-th and $j$-th sample in $A$ and $V$, respectively. Then an unbiased estimator of independence distance in the bin could be calculated as $\frac{1}{\sigma_m(\sigma_m-1)}(\sum_{i,j=1}^{\sigma_m} M_{i,j}^a M_{i,j}^v + \frac{1}{(\sigma_m-2)(\sigma_m-3)} \sum_{i,j=1}^{\sigma_m} M_{i,j}^a (M_{\cdot,\cdot}^v - 2M_{i,\cdot}^v - 2M_{\cdot,j}^v + 2M_{i,j}^v) - \frac{2}{\sigma_m-2} \sum_{i,j=1}^{\sigma_m} M_{i,j}^a (M_{i,\cdot}^v - M_{i,j}^v))$, where $M_{\cdot,\cdot}^v$, $M_{i,\cdot}^v$ and $M_{\cdot,j}^v$ denotes the row $i$ sum, row $j$ sum and grand sum, respectively of the distance matrix $M^v$. Notably, though the unbiased estimators come with a faithful expectation, we suggest running the evaluation repeatedly (e.g., 10 times) to reduce the variance of estimators.

Regardless of which statistic is employed for estimation, each $U_m$ can be considered a local test of independence within the bin $B_m$. Since $\sigma_m$ must be larger than 4 to achieve an unbiased estimator, we eventually obtain the test statistic $T = \sum_m \mathbb{1}(\sigma_m > 4)\sigma_m U_m$, and employ it as one of the evaluation metrics.

### 8.1.2 Proofs of lemma 3.2

*Assumption* 2 (Lipschitzness). For any distribution $p \in \Delta(\mathcal{V} \times \mathcal{A} \times \mathcal{Y})$, we claim it to satisfy Lipschitzness if for all $y, y' \in \mathcal{Y}$, there exists a Lipschitzness constant $L$ such that $\left\| p_{A,V|Y=y} - p_{A,V|Y=y'} \right\|_1 \leq L|y - y'|$, where $p_{A,V|Y=y}$ denotes the conditional distribution of $A, V|Y = y$ under $p$.

*Proof of (1).* The first part of the lemma gives an upper bound of $\mathbb{E}(T)$ when $p$ satisfies EOC. Assume the smoothness conditions as defined in [33], let $L$ denote the Lipschitzness constant.

Define $q_{av}(m) = \frac{\int_{B_m} p_{A,V|Y}(a,v|y)dP_Y(y)}{\mathbb{P}(Y \in B_m)} = \int_{B_m} p_{A,V|Y}(a,v|y)d\tilde{P}_Y(y)$, where $p_{A,V|Y}(a,v|y)$ is the conditional distribution of $A, V|Y = y$, and $P_Y(y)$ is the distribution of $Y$ that is absolutely continuous regarding the Lebesgue measure, and $d\tilde{P}_Y(y) = \frac{dP_Y(y)}{\mathbb{P}(Y \in B_m)}$ is the conditional distribution of $Y|Y \in B_m$. Further define $q_{a\cdot}(m) = \sum_{V \in \Delta\mathcal{V}} q_{av}(m) = \int_{B_m} p_{A|Y}(a|y)d\tilde{P}(y)$, $q_{\cdot v}(m) = \sum_{A \in \Delta\mathcal{A}} q_{av}(m) = \int_{B_m} p_{V|Y}(v|y)d\tilde{P}(y)$.

By the law of total expectation, we have $\mathbb{E}[T] = \mathbb{E}[\mathbb{E}[T|\sigma]]$, where $\sigma = (\sigma_m)_{m \in [M]}$. Since $\mathbb{E}[U_m|\sigma_m] = \sum_{a,y}(q_{av}(m) - q_{a\cdot}(m)q_{\cdot y}(m))$ is independent of $\sigma_m$, and $T = \sum_m \mathbb{1}(\sigma_m > 4)\sigma_m U_m$, then $\mathbb{E}[T] = \mathbb{E}[\mathbb{E}[T]|\sigma] = \sum_{m \in [M]} \mathbb{E}[U_m|\sigma_m]\mathbb{E}[\sigma_m \mathbb{1}(\sigma_m > 4)]$.

According to Jensen's inequality and Lipschitzness assumption, we have

$$\sum_{a,v} |q_{av}(m) - q_{a\cdot}(m)q_{\cdot y}(m)|$$

$$= \sum_{a,v} |\int (p_{A|Y}(a|y) - p_{A|Y}(a|y)d\tilde{P}_Y(y))(p_{V|Y}(v|y) - p_{V|Y}(v|y)d\tilde{P}_Y(y))d\tilde{P}_Y(y)|$$

$$\leq \int \sum_a |p_{A|Y}(a|y) - p_{A|Y}(a|y)d\tilde{P}_Y(y)| \sum_v |p_{V|Y}(v|y) - p_{V|Y}(v|y)d\tilde{P}(y)|d\tilde{P}_Y(y)$$

$$\leq \int \int \sum_a |p_{A|Y}(a|y) - p_{A|Y}(a|y')|d\tilde{P}_Y(y') \int \sum_v |p_{V|Y}(v|y) - p_{V|Y}(v|y')|d\tilde{P}(y')d\tilde{P}_Y(y)$$

$$= \int \int \left\| p_{A|Y=y} - p_{A|Y=y'} \right\|_1 d\tilde{P}_Y(y') \int \left\| p_{V|Y=y} - p_{V|Y=y'} \right\|_1 d\tilde{P}_Y(y')d\tilde{P}_Y(y)$$

$$\leq \int \int \sqrt{L|y - y'|}d\tilde{P}_Y(y') \int \sqrt{L|y - y'|}d\tilde{P}_Y(y')d\tilde{P}_Y(y)$$

$$\leq \frac{L}{d}.$$

Further, $\sum_{a,v}(q_{av}(m) - q_{a\cdot}(m)q_{\cdot y}(m)) \leq (\sum_{a,v} |q_{av}(m) - q_{a\cdot}(m)q_{\cdot y}(m)|)^2$, then $\mathbb{E}[U_m|\sigma_m] \leq \frac{L^2}{d^2}$. Since $\sum_{m \in [M]} \mathbb{E}[\sigma_m \mathbb{1}(\sigma_m > 4)] \leq \sum_{m \in [M]} \mathbb{E}[\sigma_m] = |\mathbb{D}|$. Finally we get $\mathbb{E}[T] \leq \frac{|\mathbb{D}|L^2}{d^2}$. $\square$

*Proof of (2).* The second part of the lemma gives a lower bound of $\mathbb{E}[T]$ when $d_{TV}(p, p') = \epsilon$. Let $p_m = \mathbb{P}(Y_m)$. Take two values $y, y' \in Y_m$, by triangle inequality and Lipschitzness assumption, we can get the continuity of $\left\| p_{V,A|Y=y} - p_{A|Y=y}p_{V|Y=y} \right\|_1$:

$$| \left\| p_{V,A|Y=y} - p_{A|Y=y}p_{V|Y=y} \right\|_1 - \left\| p_{V,A|Y=y'} - p_{A|Y=y'}p_{V|Y=y'} \right\|_1 |$$

$$\leq \left\| p_{V,A|Y=y} - p_{A|Y=y}p_{V|Y=y} - p_{V,A|Y=y'} + p_{A|Y=y'}p_{V|Y=y'} \right\|_1$$

$$\leq \left\| p_{V,A|Y=y} - p_{V,A|Y=y'} \right\|_1 + \left\| p_{A|Y=y'}p_{V|Y=y'} - p_{A|Y=y}p_{V|Y=y} \right\|_1$$

$$\leq \left\| p_{V,A|Y=y} - p_{V,A|Y=y'} \right\|_1 + \left\| p_{A|Y=y'} - p_{A|Y=y} \right\|_1 + \left\| p_{V|Y=y'} - p_{V|Y=y} \right\|_1$$

$$\leq 3L|y - y'|.$$

Then suppose $y_m^* \in argmax_{y \in B_m} \left\| p_{V,A|Y=y} - p_{V|Y=y}p_{A|Y=y} \right\|_1$, we have

$$\sum_{m \in [M]} \left\| p_{V,A|y=y_m^*} - p_{V|Y=y_m^*} p_{A|Y=y_m^*} \right\|_1 p_m$$

$$\geq \mathbb{E}_y \left\| p_{V,A|Y=y} - p_{V|Y=y} p_{A|Y=y} \right\|$$

$$\geq \inf_{q:EOC} \| p - q \|_1 = \epsilon.$$

By Cauchy-Schwarz inequality, we get

$$\sqrt{\sum_{v,a} (q_{va}(m) - q_{\cdot v}(m) q_{a \cdot}(m))^2} \geq \frac{\sum_{v,a} |p_{V,A|Y=y} - p_{V|Y=y} p_{A|Y=y}|}{\sqrt{|\mathcal{A}||\mathcal{V}|}}. \tag{3}$$

Further, by applying the triangle inequality and Lipschitzness assumption, the denominator of the RHS

$$\sum_{v,a} |p_{V,A|Y=y} - p_{V|Y=y} p_{A|Y=y}|$$

$$\geq \left\| p_{V,A|y=y_m^*} - p_{V|Y=y_m^*} p_{A|Y=y_m^*} \right\|_1$$

$$- \sum_{v,a} |q_{va}(m) - p_{V,A|Y}(v, a|y_m^*)|$$

$$- \sum_{v,a} |q_{a \cdot}(m)(q_{\cdot v}(m) - p_{V|Y}(v|y_m^*)|$$

$$- \sum_{v,a} |(p_{V|Y}(v|y_m^*))(q_{a \cdot}(m) - p_{A|Y}(a|y_m^*))|.$$

The last three items are less than $\frac{L}{d}$, then

$$\sum_{v,a} |p_{V,A|Y=y} - p_{V|Y=y} p_{A|Y=y}| \geq \left\| p_{V,A|y=y_m^*} - p_{V|Y=y_m^*} p_{A|Y=y_m^*} \right\|_1 - \frac{3L}{d}. \tag{4}$$

Combining Eqs (3) and (4), we get $\sqrt{\sum_{v,a} (q_{va}(m) - q_{\cdot v}(m) q_{a \cdot}(m))^2} \geq$ $\frac{\left\| p_{V,A|y=y_m^*} - p_{V|Y=y_m^*} p_{A|Y=y_m^*} \right\|_1 - \frac{3L}{d}}{\sqrt{|\mathcal{A}||\mathcal{V}|}}.$

By summing over all $m$, we have

$$\sum_{m \in [M]} \sqrt{\mathbb{E}[U_m|\sigma_m]} p_m$$

$$= \sum_{m \in [M]} \sqrt{\sum_{v,a} (q_{va}(m) - q_{\cdot v}(m) q_{a \cdot}(m))^2} p_m$$

$$\geq \sum_{m \in [M]} \frac{\left\| p_{V,A|y=y_m^*} - p_{V|Y=y_m^*} p_{A|Y=y_m^*} \right\|_1 - \frac{3L}{d}}{\sqrt{|\mathcal{A}||\mathcal{V}|}} p_m$$

$$\geq \frac{\epsilon - 3\frac{L}{d}}{\sqrt{|\mathcal{A}||\mathcal{V}|}}.$$

Assume that each $B_m$ has sufficient samples such that $\sigma_m > 4$, $\mathbb{E}[T] = \mathbb{E}[\mathbb{E}[T]|\sigma] = \sum_{m \in [M]} \mathbb{E}[U_m|\sigma_m] \mathbb{E}[\sigma_m \mathbb{1}(\sigma_m > 4)] = \sum_{m \in [M]} \mathbb{E}[U_m|\sigma_m] \mathbb{E}[\sigma_m] = \sum_{m \in [M]} \mathbb{E}[U_m|\sigma_m] |\mathbb{D}| p_m$. By Cauchy-Schwarz inequality, we have the following:

$$\mathbb{E}[T] = \sum_{m \in [M]} \mathbb{E}[U_m|\sigma_m] |\mathbb{D}| p_m \geq \frac{(\sum_{m \in [M]} \sqrt{\mathbb{E}[U_m|\sigma_m]} |\mathbb{D}| p_m)^2}{\sum_{m \in [M]} |\mathbb{D}| p_m} \geq \frac{|\mathbb{D}|(\epsilon - 3\frac{L}{d})^2}{4|\mathcal{A}||\mathcal{V}|}. \tag{5}$$

$\square$

## 8.2 Details and Proofs for the Proposed Approach BFQR (Section 4)

### 8.2.1 Pseudocode for Algorithm 1

The pseudocode for the main component of BFQR is detailed in Algorithm 1.

---

**Algorithm 1** Binned Fair Quantile Regression (BFQR)

---

**Input:** Training Data $D_{tr}$, Calibration Data $D_c$, Test Data $X_t$ with sensitive attributes $A_t$, Desired error rate $\alpha$.

1: Train quantile regression models $\hat{q}_{\alpha_{lo}}$ and $\hat{q}_{\alpha_{hi}}$ on $D_{tr}$, $\alpha_{lo} = \alpha/2$ and $\alpha_{hi} = 1 - \alpha/2$.
2: **for all** $i \in \mathbb{D}_c$ **do**
3:     Fit $\hat{q}_{\alpha_{lo}}$ and $\hat{q}_{\alpha_{hi}}$ on $D_c$, get prediction intervals $\hat{C}(X_i) = [\hat{q}_{\alpha_{lo}}(X_i), \hat{q}_{\alpha_{hi}}(X_i)]$.
4:     Compute conformity scores $R(X_i, Y_i) = max\{\hat{q}_{\alpha_{lo}}(X_i) - Y_i, Y_i - \hat{q}_{\alpha_{hi}}(X_i)\}$.
5: **end for**
6: Divide $[Y_{min}, Y_{max}]$ into $M$ equal-mass bins, $B_1, ..., B_M$.
7: Call Algorithm 2 to optimized $\beta_m$, $m \in \{1, \ldots, M\}$.
8: **for all** $m \in \{1, \ldots, M\}$ **do**
9:     **for all** $a \in A$ **do**
10:         Compute quantile value $G_{a,m}(\beta_m)$.
11:     **end for**
12: **end for**
13: **for all** $a \in A$ **do**
14:     **for all** $i \in D_c(a) = \{i : i \in \mathbb{D}_c, A_i = a\}$ **do**
15:         **for all** $m \in \{1, \ldots, M\}$ **do**
16:             Assume the true label $Y_i \in B_m$, compute corresponding prediction interval $C_m(X_i) = B_m \cap [\hat{q}_{\alpha_{lo}}(X_i) - G_{a,m}(\beta_m), \hat{q}_{\alpha_{hi}}(X_i) + G_{a,m}(\beta_m)]$.
17:         **end for**
18:         $C(X_i) = \bigcup_m C_m(X_i)$
19:     **end for**
20: **end for**

**Output:** Conformalized Prediction Interval $C(X_t)$.

---

### 8.2.2 Proofs

*Proof of Theorem 4.1.* According to Assumption 1 about exchangeability, for any sensitive group $a$ and calibration bin $B_m$, the conformity scores $S$ for all samples in this group $\mathbb{D}_c(a, m)$ are exchangeable. Adding the score of new test data within the same group would not change the distribution of the scores in this group, then exchangeability still holds. With the additional assumption that conformity scores are almost surely distinct [20, 28, 59, 35, 11], we get the following:

$$\beta_m \leq \mathbb{P}\{S_{n+1} \leq G_{a,m}(\beta_m)|A_{n+1} = a, Y_{n+1} \in B_m\}$$
$$= \mathbb{P}\{Y_{n+1} \in C(X_{n+1})|A_{n+1} = a, Y_{n+1} \in B_m\}$$
$$\leq \beta_m + 1/(|\mathbb{D}_c(a, m)| + 1).$$

Hence, $\beta_m \leq \mathbb{E}[Y_{n+1} \in C(X_{n+1})|A_{n+1} = a, Y_{n+1} \in B_m] \leq \beta_m + 1/(|\mathbb{D}_c(a, m)| + 1)$. In the worst case, there exists a group such that the expectation equals $\beta_m$ whilst the expectation for the group with the least samples equals $\beta_m + 1/(|\mathbb{D}_c(a, m)| + 1)$. Therefore, the expectation of max coverage gap inside the $m$-th bin is upper bounded by $max_a\{1/(|\mathbb{D}_c(a, m)| + 1)\}$. $\qquad\square$

*Proof of Theorem 4.2.* According to the law of total probability and Theorem 4.1, we have

$$\mathbb{P}\{Y_{n+1} \in C(X_{n+1})\}$$
$$= \sum_a \sum_m \mathbb{P}\{Y_{n+1} \in C(X_{n+1})|A_{n+1} = a, Y_{n+1} \in B_m\}\mathbb{P}\{A_{n+1} = a, Y_{n+1} \in B_m\}$$
$$\geq \sum_a \sum_m \beta_m\mathbb{P}\{A_{n+1} = a, Y_{n+1} \in B_m\} = \beta_m.$$

$\qquad\square$

### 8.3 Details for the Constrained Optimization (Section 4.3)

#### 8.3.1 Detailed Methods

A feasible solution to Equation (1) can be quickly found. Recall split conformal prediction introduced in Section 2, $\hat{Q}_{1-\alpha}(S, \mathbb{D}_c)$ denotes the $1 - \alpha$ quantile value of scores in $\mathbb{D}_c$. Adjust $\hat{C}(X_i)$, $X_i \in \mathbb{D}_c$ by $\hat{Q}_{1-\alpha}(S, \mathbb{D}_c)$. For example, let $C^S(X_i) = [\hat{q}_{\alpha_{lo}}(X_i) - \hat{Q}_{1-\alpha}(S, \mathbb{D}_c), \hat{q}_{\alpha_{hi}}(X_i) + \hat{Q}_{1-\alpha}(S, \mathbb{D}_c)]$, then $C^S(X_i)$, $X_i \in \mathbb{D}_c$ naturally has coverage of $1 - \alpha$. Compute the coverage rate $\beta_m$ of adjusted calibration prediction sets in each bin $B_m$, we get a feasible solution which we use as a starting point of the optimization process.

For any $i$, let $m_i^-$ and $m_i^+$ denote the number of the first bin and last bin that has a non-null intersection $C_m(X_i)$. In the relaxed dummy prediction set $C^d(X_i)$, $C_m(X_i)$ are substituted by $B_m$ for $m_i^- < m < m_i^+$. Naturally, the width of continuous prediction interval $C^d(X_i)$ is an upper bound of the original $C(X_i)$, as all the possible disjoint gaps are filled.

Denote $W$ to be the objective function in Eq. 1, i.e., the mean width of the prediction intervals on test data. Using the dummy prediction interval $C^d(X_i)$ as a bridge, and let $W^S = \sum_{i \in |\mathbb{D}_t|} |C^S(X_i)|/|\mathbb{D}_t|$ denote the mean width of prediction intervals at the starting point, we provide an upper bound of the mean width of our prediction intervals $W$:

*Proposition* 1. $W \leq W^S + \sum_{i \in |\mathbb{D}_t|} [G_{A_i, m_i^-}(\beta_{m_i^-}) + G_{A_i, m_i^+}(\beta_{m_i^+}) - 2\hat{Q}_{1-\alpha}(S, \mathbb{D}_c(a))]/|\mathbb{D}_t|$.

Though easier to compute, the RHS of 1 is still a combination of quantile functions, and the number of first and last intersecting bins $m_i^-$ and $m_i^+$ are related with $G_{A_i, m}(\beta_m)$, meaning they vary with the change of $\beta_m$. Therefore, we propose several good properties that allow us to approximate subgradients of quantile functions. Denote $Q_{a, m}(\beta)$ as the underlying distribution for the quantile function of scores on the calibration data $\mathbb{D}_c(a, m)$, which is what $G_{a, m}(\beta)$ estimates for.

**Lemma 8.1.** *For an exchangeable sequence of random variables, almost surely, the difference between the empirical and the predictive distribution functions converges to zero uniformly. [40]*

*Proposition* 2. For any $0 < \beta < 1$, there exists a $\epsilon$ such that $|Q_{a, m}(\beta) - G_{a, m}(\beta)| \leq \epsilon$. Then for any $0 < \beta_1 < \beta_2 < 1$, the maximal absolute difference between the slope of true quantile function $t_{a, m} = \frac{Q_{a, m}(\beta_1) - Q_{a, m}(\beta_2)}{\beta_1 - \beta_2}$ and the slope of empirical quantile function $\hat{t}_{a, m} = \frac{G_{a, m}(\beta_1) - G_{a, m}(\beta_2)}{\beta_1 - \beta_2}$ is upper bounded by $\frac{2\epsilon}{\beta_1 - \beta_2}$. This difference converges to 0 almost surely when $|\mathbb{D}_c(a, m)| \to \infty$.

The dummy prediction interval $C^d(X_i)$ also exhibits some properties that can be further utilized to facilitate computation. First, any change of $\beta_m$ for bins whose $m$ is between $m^-$ and $m_i^+$ would not change $C^d(X_i)$. Second, for any increase in $\beta_m$, the length of the intersection $|C_m(X_i)|$ in bin $B_m$ for any $X_i$ can not decrease, there are three cases: 1) $|C_m(X_i)|$ stays at 0 or $Y_m^+ - Y_m^-$, meaning still no intersection between prediction interval $[\hat{q}_{\alpha_{lo}}(X_i) - G_{a, m}(\beta_m), \hat{q}_{\alpha_{hi}}(X_i) + G_{a, m}(\beta_m)]$ and bin $B_m$, or the prediction interval are all covered in bin, therefore, the true change in interval lengths is 0; 2) $|C_m(X_i)|$ increases from 0 to any number within $(0, Y_m^+ - Y_m^-)$, or from any number within $(0, Y_m^+ - Y_m^-)$ to all covered in bin; 3) $|C_m(X_i)|$ increases from one number to another within $(0, Y_m^+ - Y_m^-)$. For the first two cases, our estimation $G_{a, m}(\beta_1) - G_{a, m}(\beta_2)$ exceeds the true value, and for the last one, our estimation is exact. A similar property holds for the decrease in $\beta_m$.

According to the above properties, we maintain a dictionary $D$ that stores the number of data that have a non-full intersection in each bin, denoted as $u_m$, which includes case 3 and part of case 2. We also deliberately select steps at each iteration at a small value that not exceeds $1/|\mathbb{D}_c(a, m)|$, and we allow estimation discrepancies in the algorithm, then case 2 has very little influence that could be ignored. Therefore, we use $grad_m^+ = u_m \cdot \hat{t}_m^+/|\mathbb{D}_m|$ and $grad_m^- = u_m \cdot \hat{t}_m^-/|\mathbb{D}_m|$ as the approximated increasing and decreasing gradient for each bin $B_m$.

#### 8.3.2 Pseudocode for Algorithm 2

The pseudocode for the constrained optimization is detailed in Algorithm 2.

**Algorithm 2** Robust Optimization Method with EOC Constraint

---

**Input:** Bins $B_1, ..., B_M$, Prediction intervals $\hat{C}(X_i) = [\hat{q}_{\alpha_{lo}}(X_i), \hat{q}_{\alpha_{hi}}(X_i)]$ and Scores $R(X_i, Y_i)$ for all $i \in \mathbb{D}_c$, Desired coverage rate $1 - \alpha$, Max iteration round $R$, Allowed Estimation Error $\epsilon$.

1: Compute $\hat{Q}_{1-\alpha}(S, \mathbb{D}_c)$ as the $1 - \alpha$ quantile of $\{R(X_i, Y_i), i \in \mathbb{D}_c\}$.
2: Compute $\tilde{C}(X_i) = [\hat{q}_{\alpha_{lo}}(X_i) - \hat{Q}_{1-\alpha}(S, \mathbb{D}_c), \hat{q}_{\alpha_{hi}}(X_i) + \hat{Q}_{1-\alpha}(S, \mathbb{D}_c)]$ for all $i \in \mathbb{D}_c$.
3: **for all** $m \in \{1, \ldots, M\}$ **do**
4:     Compute the coverage rate of $\tilde{C}(X_i)$ for $i \in D_c(m)$ as initial value $\beta_m^0$.
5: **end for**
6: Initiate dictionary $D$ that stores the number of data that have a non-full intersection in each bin with initial values $\{\beta_m^0, m \in 1, \ldots, M\}$, and $r \leftarrow 0$.
7: Initiate Array $A$ of shape $2 * A * M$ that stores the max possible step.
8: **while** $r < R$ **do**
9:     **for all** $m \in \{1, \ldots, M\}$ **do**
10:         Compute $grad_m^+$ and $grad_m^-$, along with max step $\eta_m^+$ and $\eta_m^-$.
11:     **end for**
12:     $m_{min}^+ \leftarrow \arg\min m : grad_m^+, m_{max}^+ \leftarrow \arg\max m : grad_m^-$.
13:     $\eta^r = \min\{\eta_m^+, \eta_m^-\}$
14:     **if** $m_{min}^+ + 2\epsilon/\eta^r > m_{max}^-$ **then**
15:         Break.
16:     **end if**
17:     Update $\beta_{m_{min}^+}^{r+1} \leftarrow \beta_{m_{min}^+}^r + \eta^r$, and $\beta_{m_{max}^-}^{r+1} \leftarrow \beta_{m_{max}^-}^r - \eta^r$.
18:     Update dictionary $D$ and Array $A$.
19: **end while**
**Output:** Optimal coverage rate $\beta_m^r$ for each bin $m$.

---

## 8.4 Experimental Details

Our proposed algorithm is highly computationally efficient. All experiments are conducted on Google Colab with only CPUs.

### 8.4.1 Synthetic Data Generation Process

The data generation process for the synthetic experiments is as follows:

$$
Y = \begin{cases} (A + \sum_{i=1}^{10} X_i + 10\epsilon_1)\epsilon_3, & A = 0, \quad 0 \le \epsilon_4 \le 0.1, \\ 10A\epsilon_2, & A = 1, \quad 0.1 < \epsilon_4 \le 0.3, \\ (A + \sum_{i=1}^{10} X_i + 10\epsilon_1)\epsilon_3, & A = 2, \quad 0.3 < \epsilon_4 \le 1, \end{cases} \quad (6)
$$

where $\epsilon_1, \epsilon_2, \epsilon_3, \epsilon_4 \sim \mathcal{N}(0, 1)$, $X_i \sim \exp(1)$.

### 8.4.2 Convergence Analysis of BFQR

We experiment on the Adult dataset to analyze the convergence of our algorithm with five different random seeds for data splitting. The convergence process of the width of prediction intervals $(W)$ is depicted by solid lines, while the convergence of the dummy upper bound with continuous prediction intervals $(W^S)$ is represented by dotted lines, as shown in Figure 4.

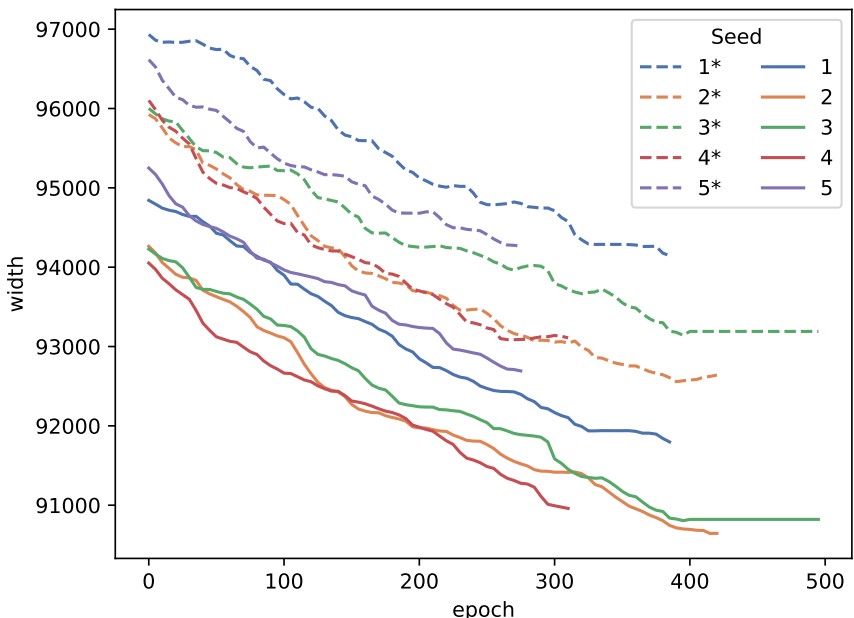

Figure 4: Converging process of prediction interval width. The number in the legend denotes the seed selected for splitting data. The dotted lines with '*' in the legend show the process for continuous prediction intervals, while the solid lines show the process for disjoint prediction intervals.

The convergence analysis results demonstrate that both $W^S$ and $W$ exhibit similar convergence trends. This indicates that optimizing $W^S$ leads to a reduction in $W$, thereby validating Proposition 1. Furthermore, the continuous decrease in $W$ highlights the efficiency and effectiveness of the multiple approximation techniques used in Sec. 8.3.1. These techniques enable faster computation and preserve the information of true gradients.