# OpenReview forum: "Equal Opportunity of Coverage in Fair Regression"
_NeurIPS.cc/2023/Conference — NeurIPS 2023 poster_

### Official Review · Reviewer_FNuJ · 2023-06-08

**Soundness:** 3 good
**Presentation:** 3 good
**Contribution:** 2 fair
**Rating:** 4
**Confidence:** 4

**Summary:**

Authors propose a new uncertainty-aware fairness – Equal Opportunity of Coverage (EOC) since current definition do not guarantee equal coverage rates across more fine-grained groups (e.g., low-income females) conditioning on the true label and is biased in the assessment of uncertainty.

**Strengths:**

- paper is technically sound
- experimental results are promising

**Weaknesses:**

- presentation is mostly verbose and unclear
- novelty is limited

**Questions:**

The paper is, in my opinion, technically sound with promisin gresults.
Nevertheless I think that the approach is not completely novel.
Many works in the literature of fairness focus on distribution matching (see also NeurIPS papers when they use MMD, Sinkhorn distance, or first order matching) and there in no discussion on the relation with these works and a comparison with the idea of this work.
Authors should clarify and elaborate on this.

**Limitations:**

Authors should better clarify the novelty of the work and its impact.

---

> ### Author Rebuttal · Authors · 2023-08-10
>
> Thank you for your suggestions. We study fair regression under uncertainty in this paper. Our major contributions are summarized as follows:
>
> We identified several important limitations in prior works: equal coverage is not guaranteed when conditioned on different values of $Y$ and; interval width is largely increased (i.e., low prediction confidence) when ensuring equal coverage.
>
> To tackle these limitations, (1) we proposed a novel uncertainty-aware fairness notion, EOC. As mentioned by Reviewer RXWV, the definition of EOC (a variant of equal opportunity) was not discussed in any other works, and it introduced a fresh perspective to the field. (2) We also proposed a novel effective and efficient method that can improve EOC meanwhile reduce interval width. According to Reviewer 1ZNj, achieving equal opportunity in regression is less researched but is an important problem. Our work reflects our efforts to tackle challenges in this setting.
>
> Regarding the concerns about using distribution matching techniques, we need to clarify that distribution matching is only used for evaluation, and should not be considered as a contribution of this work.
> Our proposed post-processing method is distinct from in-processing fairness methods, where distribution matching is commonly employed. While adapting our method to in-processing settings is very interesting, it is required more advanced techniques which is beyond the scope of this work. Therefore, we did not compare our method with models using a distribution matching framework.

---

> > ### Comment · Reviewer_FNuJ · 2023-08-15
> >
> > I think that the answer of authors do not fully address the point raised.
> > E.g., [1] or [2] are distribution matching post-processing techniques.
> > I will keep my score as it is.
> >
> > [1] Chzhen, E. et al. Fair Regression with Wasserstein Barycenters, NeurIPS, 2020.
> > [2] Ray, J. et al. Wasserstein fair classification, UAI, 2020.

---

> > > ### Author Response · Authors · 2023-08-16
> > > **Comparison with distribution matching post-processing techniques in interval prediction**
> > >
> > > We appreciate your attention to these papers [1,2]. Our current study focuses on the concept of equal opportunity of coverage (EOC), which pertains to fairness in interval prediction. However, it's important to note that the approach presented in [1,2] isn't directly suitable for achieving EOC, as its design revolves around predicting values while adhering to a demographic parity (DP) constraint.
> > >
> > > In contrast, our research involves a comparison with CFQR [3], also referred to as [16] in our paper. CFQR is an adaptation of the method introduced in [1] for our interval prediction context. CFQR modifies fair regression with Wasserstein Barycenters for interval prediction through two key steps: 1) generating prediction intervals at a specified coverage rate and 2) ensuring that the upper and lower bounds of these intervals satisfy the DP constraint. To attain the first objective, CFQR employs a conformal prediction framework similar to ours, while drawing from the theories and techniques outlined in [1] to achieve DP for interval bounds.
> > >
> > > The outcomes presented in Tables 1-3 of our paper demonstrate that CFQR is comparatively less effective than our proposed approach. In certain datasets, our method showcases improved EOC. Moreover, when compared to our method, CFQR's excessively wide prediction intervals reduce their informativeness, and the coverage rates across groups of CFQR are less equalized.
> > >
> > > [1] Chzhen, E., Denis, C., Hebiri, M., Oneto, L., \& Pontil, M. (2020). Fair regression with wasserstein barycenters. Advances in Neural Information Processing Systems, 33, 7321-7331.
> > >
> > > [2] Jiang, R., Pacchiano, A., Stepleton, T., Jiang, H., \& Chiappa, S. (2020, August). Wasserstein fair classification. In Uncertainty in artificial intelligence (pp. 862-872). PMLR.
> > >
> > > [3] Liu, Meichen, et al. "Conformalized Fairness via Quantile Regression." Advances in Neural Information Processing Systems 35 (2022): 11561-11572.

---

### Official Review · Reviewer_1ZNj · 2023-07-04

**Soundness:** 4 excellent
**Presentation:** 3 good
**Contribution:** 3 good
**Rating:** 7
**Confidence:** 4

**Summary:**

This paper studies fair machine learning under predictive uncertainty.
They propose a new notion of fairness named Equal Opportunity of Coverage (EOC), building upon "equalized coverage." While EC ensures that every demographic group receives the same level of prediction certainty, EOC ensures that every demographic group with similar true labels receives the same level of prediction certainty.
This extension is similar to the extension of equal opportunity from statistical parity.
With EOC, they also ensure that the entire population's coverage can be maintained at a predetermined level.
To achieve EOC, they propose a post-processing method called Binned Fair Quantile Regression (BFQR), a distribution-free post-processing method for a given trained model.

They show the power of their method both on the simulated data and real-world datasets.


**Strengths:**

This is a very interesting research direction.

Fair regression is also a challenging problem, and little research extends equal opportunity (and its variants) to regression settings.
This is a promising research direction.


**Weaknesses:**

I have a few apprehensions regarding the notation and the inadequate introduction of certain mathematical objects. Please see my comments below.


**Questions:**

### Major Comments:

1) Line 83: Why are the calibration and test sets exchangeable?
2) Line 111: How does this indicate "V \perp A"?
3) TV definition in Line 122 needs to be more rigorous. Please define what $p$ and $p'$, which space they live, etc.
4) Up to Line 135, I need clarification on $p$. Is the true data-generating distribution, or is it the empirical distribution?
5) In Lemma 3.2, what is the sum of independence measures? What is the exact reference of Lemma 3.2 in [30]?
6) Why do you choose to use TV? (I understand why not KS distance, but I do not understand why you choose TV over other metrics. An example could be Wasserstein distance, MMD, KL divergence etc.)
7) Line 164: The coverage number $\beta_m$ is the same for each group, right? It is different across subgroups. Then, I do not understand why $\beta$ is computed separately for each group. Isn't it computed separately for each bin? I do not understand that part.
8) The difference between BFQR and BFQR* is not discussed in the main text. What are the differences between the two?

9) What are the conditions that Assumption 1 is violated? How strong is this assumption?

10) Is it possible to include EOC constraints somehow during training?
Do you think it would be a stronger method if this method could be an in-processing method?



### Minor Comments:

1) Typo Line 85: "Pr(" instead of "P(".
2) In (1), $min$ and $s.t.$ should be aligned, and "s.t." should be a math operator.
3) Please introduce each object rigorously. It is unclear to me what sort of mathematical objects you are referring to until I see them used. For example, quantile regression. What is $\hat q$? Is this a function? Is it a vector? What is $\alpha_{lo}$? This would help the readers a lot if you were to introduce each object appearing in the paper rigorously.
4) Line 252: "l" should be capitalized in lemma
5) Line 312:  "t" should be capitalized in theorem
6) Please indicate the best results in the experiment with bold font.
7) Line 224: "W" should not be capitalized.
8) Line 360: "\beta_m" instead of $\beta m$
9) Line 272: "Section" instead of "Sec." (also 290).
10) Line 311: "Figure" instead of "Fig."
11) What is the probability distribution $\mathbb P$ appeared suddenly in Theorem 4.1 and Theorem 4.2?
I do not think the notation of this paper is unified. Please unify the notation during the rebuttal. Defining notation rigorously and sticking with it throughout the paper is important. Otherwise, the paper becomes unnecessarily hard to follow.
12) I think it would be safe to drop the "under no label distribution shift on test data" assumption from Theorem 4.2. In the beginning, if you were to define the true data-generating distribution and explain how the test data and the calibration data are created from iid samples from this true data-generating distribution, then there is no need to mention this shift. If you could cover the shifted distribution case, then I would suggest mentioning it.


**Limitations:**

Yes.

---

> ### Author Rebuttal · Authors · 2023-08-10
>
> We sincerely appreciate your thoughtful and constructive suggestions which help enhance the clarity and accessibility of this paper. Below are our replies to your questions/comments, and we will revise our ambiguous expressions to avoid confusion and misunderstanding.
>
> 1. The exchangeability of calibration and test sets is the fundamental assumption of classic conformal prediction to guarantee its coverage.
>
> 2. Symbol '⫫' means independent of in our paper.
>
> 3.
> Define distribution $p, p' \in \Delta(\mathcal{V} \times \mathcal{A} \times \mathcal{Y})$, and $\mathcal{V}$, $\mathcal{A}$, $\mathcal{Y}$ is the domain of $V$, $A$, $Y$ respectively, as shown in the preliminary.
> The total variation distance between $p$ and $p'$ is defined as $d_{TV}(p,p') := \sup_{S \subseteq \Delta(\mathcal{V} \times \mathcal{A} \times \mathcal{Y})} $\{$ p(S) - p'(S)\$\}$ = \frac{1}{2} \sum_{x \in \Delta(\mathcal{V} \times \mathcal{A} \times \mathcal{Y})} |p(x) - p'(x)|$.
> We further denote $P_{EOC}$ as the property of EOC, the set of all distributions in defined space that satisfy EOC, i.e., $P_{EOC}:= $\{$p \in \Delta(\mathcal{V} \times \mathcal{A} \times \mathcal{Y}): if (V, A, Y) \in p, V ⫫ A | Y $\}. Fix distribution $p' \in P_{EOC}$ as of minimum TV distance to $p$, i.e., for all distributions $\forall q \in P_{EOC}$, $d_{TV}(p, p') \leq d_{TV}(p, q)$.
>
> 4. $p$ is the true data-generating distribution.
>
> 5. The details of the independence measure can be found in Appendix 8.1.1. We are inspired by Theorem 5.2 in the referred paper [30] and simplify its notation.
>
> 6. We follow referred works [29,30] in the paper using TV distance as it is a commonly used metric for measuring how far two variables are from being conditionally independent. TV distance is a more appropriate metric in this work as it is symmetric compared with KL divergence and easier to compute than Wasserstein distance. Since MMD is also widely used for conditional independence tests [1], we conjecture that it would produce similar evaluation results.
>
> 7. For better understanding, we illustrate the process described in Lines 161 -171 with Figure 3 in the attached pdf file for the global response. We will include it in the next version of this work.
>
> 8. We apologize for the confusion. The difference between BFQR and BFQR* is described in the experiment setup section (Lines 270-272). BFQR* is the convex hull of intervals in BFQR (Line 230), disabling disjoint intervals. This variant of BFQR is meaningful in some real-world settings. For example, for MEPS data, predicting a healthcare system utilization score to be in [0,1], [4,5], or [10,15] is confusing. Although the total interval width is small, survey researchers cannot decide whether the score is high or low. In this case, a continuous [0,15] with a wider interval makes more sense.
>
> 9. Assumption 1 assumes the exchangeability of calibration and test data with the same bin and is a relaxation of the i.i.d. assumption. It does not impose assumptions on data distribution and base models. It is violated when there is a distribution shift of conformity scores within a bin from calibration to test data. Relaxing the exchangeability assumption is important, however, it needs more advanced techniques which are beyond the scope of this work. We will explore potential solutions (e.g., sample reweighting) in future works.
>
> 10. We agree that post-processing methods typically underperform in-processing methods. However, there are many situations where in-processing methods are not applicable, e.g., the prediction model is a pretrained black-box regression model or a flexible and computationally efficient method is required. Moreover, conformal prediction itself is a post-hoc uncertainty quantification approach. Therefore, it is a natural application to post-processing methods. Adapting our methods to an in-processing setting will be considered in future works.
>
> We have carefully reviewed your minor comments and will revise all the typos and format problems. We clarify the other ambiguities that require further explanations (numbered in correspondence with your comments):
>
> 3. Let $\\hat{q}_{\\alpha}$ denote the $\\alpha$-th conditional quantile regression function, i.e., for $i$-th sample $(X_i, Y_i)$, (here are some unknown errors with Markdown)
>
> $\hat{q}_{\alpha}(X_i) := \inf $\{$ y\in \Delta \mathcal{Y}: Pr$\{$Y_i \leq y | X = X_i $\}$ \geq \alpha $\}.
>
> Fix the lower and upper quantiles as $\alpha_{lo} = \alpha/2$ and $\alpha_{hi} = 1 - \alpha/2$, then $\\hat{q}_{\\alpha_{lo}}(X_i)$
> and $\\hat{q}_{\\alpha_{hi}}(X_i)$ denote lower and upper quantile regression functions, respectively.
>
> 11. Thank you for the reminder about the notion inconsistency problem. We will use $Pr$ in this paper.
>
> 12. We remove the "under no label distribution shift on test data" assumption. Thanks for pointing it out!
>
> [1] Gretton, A., Borgwardt, K. M., Rasch, M. J., Schölkopf, B., & Smola, A. (2012). A kernel two-sample test. The Journal of Machine Learning Research, 13(1), 723-773.

---

> > ### Comment · Reviewer_1ZNj · 2023-08-20
> >
> > Thank you very much for your response! I will keep my score.

---

### Official Review · Reviewer_RXWV · 2023-07-06

**Soundness:** 3 good
**Presentation:** 2 fair
**Contribution:** 3 good
**Rating:** 4
**Confidence:** 3

**Summary:**

The authors propose a method to achieve same coverage rates between different groups conditioned on similar targets in addition to satisfying a marginal coverage rate on the entire population. The proposed objective is denoted as Equal Opportunity of Coverage (EOC) and the authors propose a two step approach denoted as Binned Fair Quantile Regression (BFQR) to address it. This method partitions the target space into bins and for each bin finds a coverage rate such that EOC is satisfied (i.e., equal coverage across sensitive groups inside the bin and global coverage rate level is achieved.). At test time the prediction intervals are built as the union of the intersections of each bin and its associated conformal prediction interval taking into account the group membership of the test sample.

**Strengths:**

I think the problem presented in this work is well motivated. Figure 1 is a nice graphical example to motivate the importance of considering EOC over Equalized coverage.

The definition of EOC is clear and as far as I know it was not discussed in other works. Its associated theory is well linked to previous works on Equal Opportunity.

Empirically, the proposed BFQR method is improving the maximum coverage gap between groups across the bins when compared to almost all the competing approaches. It also seems to achieve similar group coverage (average) across the groups and maintaining smaller interval widths.

**Weaknesses:**

I think the clarity of the presentation should be improved. In particular, I found that the description of the proposed procedure in Section 4 needs better clarification. For instance, L161 to L164 mentions that to enforce EOC the coverage rate within each bin $\beta_m$ is the same across groups. The method to achieve this in practice is not clear to me when reading the main manuscript. Paragraph in L116 discusses that TV distance w.r.t. the closest EOC distribution (p’) is measured. However, it is not clear enough how p’ is obtained and how this distance is obtained. I understand the authors are citing the works from [39,40] where an ‘easy-to'-compute statistic T’ is being used but this is an important point in the proposed method and should be clarified here for reproducibility purposes.

I found that Algorithm 1 in Appendix is a better summary of the description provided in last part of Section 4.1 and 4.2. However, this Algorithm builds on top of the previously estimated bin coverage rate $\beta_m$.

I think the organization of the paper is a bit convoluted since Section 4.3 describes how to obtain $\beta_m$ but this was already discussed in Section 4.1 and Section 3. Moreover, Section 4.1 and Section 4.2 were relying on this. I think that the description on how $\beta_m$ is obtained could be improved, it could be made more concise and informative.

Figure 1 provides a nice and intuitive illustration of the problem, it would be nice to see the same plot after the proposed post processing procedure is applied in comparison with the other competing methods.

From Algorithm 1 in the Appendix and L172-173 it seems that during test time the sensitive attribute needs to be available. I can think of this as a weak point in the fairness setting since the sensitive attribute needs to be available at test time. However, this is also the case for other existing fairness post-processing approaches.

**Questions:**

If the sensitive attribute needs to be disclosed for each test sample, I think a reasonable comparison would be against a model that performs a group conditional split conformal approach that conditions on the sensitive attribute and a binning based on the outcomes of the quantile regression model. Is any of the competing approaches doing this?

CFQR seem to be the method achieving the best Max coverage gap in Table 1 and Table 3 by a large margin. In particular, in Table 1 the max coverage gap is 0.19 but group A=1 has an overall coverage of 71.38 versus the other two groups whose coverage are larger than 93. I wonder if this is because across the majority of bins group A=1 is the one with the worse coverage even though by a small amount per bin (i.e., less than 0.19). How many bins where considered to the synthetic example?  I think to achieve that large difference between group coverages with small coverage gaps there has to be many bins.

**Limitations:**

yes

---

> ### Author Rebuttal · Authors · 2023-08-10
>
> We appreciate your valuable and informative suggestions and would clarify the questions and concerns in the following.
>
> **Q1**: Unclear writing in Section 4.
>
> **A1**: We will rephrase this paragraph and add a new Figure 3 illustrating how $\beta_m$ is enforced to be equal within the same bin (see attached pdf in the global response). While our method relies on the standard exchangeability assumption (Assumption 1) in conformal prediction, the distribution of conformity scores can be different across different groups within the same bin $m$.
> Therefore, to guarantee coverage rates for all groups in bin $m$ to be the same ($\beta_m$), we calibrate the test data with the $\beta_m$-th quantile of conformity scores calculated separately within each group.
>
> Here is a formal definition of $p'$ in L116: We denote $P_{EOC}$ as the property of EOC, the set of all distributions in defined space that satisfy EOC, i.e., $P_{EOC}:= $\{ $p \in \Delta(\mathcal{V} \times \mathcal{A} \times \mathcal{Y}): \text{if}  (V, A, Y) \in p, V⫫A | Y $ \}. $p' \in P_{EOC}$ is the distribution with minimum TV distance to $p$, i.e., for all distributions $\forall q \in P_{EOC}$, $d_{TV}(p, p') \leq d_{TV}(p, q)$. We will add the definition to the next version of this work.
>
> To avoid introducing complex definitions and notations, we intuitively introduce $T$ as the sum of the independence measure with bins in the paper. The details of the computation of $T$ can be found in Appendix 8.1.1, where we follow Section 4.1 in the referred paper [30]. Detailed implementation of codes is also available in Line 61. We will also add a more elaborate description to the next version of this work.
>
> **Q2**: Section 4 can be better organized.
>
> **A2**: Section 4 is organized as follows: we first enforce the independence of $A$ and $V$ within each interval of $Y$ (i.e., EOC) for the calibration data in Section 4.1, and then leverage conformal prediction to achieve EOC for test data in Section 4.2. Lastly in Section 4.3, we describe an efficient and robust optimization approach that optimizes both EOC and widths of prediction intervals. We organize Section 4.3 behind Sections 4.1 and 4.2 because, in Section 4, the most important aspect of the proposed method is its ability to improve EOC (Sections 4.1 and 4.2). On this premise, we then expect tighter intervals (Section 4.3). We concur that it is natural to elaborate on how to improve $\beta_m$ right after introducing it in Section 4.1, but it may arise other difficulties, e.g., the objective function in optimization is derived from Section 4.2.
>
> **Q3**: Add plots for competing methods.
>
> **A3**: Thanks for the suggestion. We added Figure 4 to clearly present the comparison results in the pdf attached to the global response. We will add this figure to the next version of this draft.
>
> **Q4**: Missing sensitive attributes.
>
> **A4**: As one of the first few works studying equalized coverage, this work as well as the baselines adopts conventional settings in fair machine learning, which assumes the availability of sensitive attributes in the test phase.
> For practical concerns, it is surely a promising future direction to adapt our method to handle missing sensitive attributes. We will discuss this limitation in the Discussion section.
>
> **Q5**: Comparison with the conditional split conformal inference approach.
>
> **A5**: We compared our method with Group-conditional Conformalized Quantile Regression (GCQR) in the paper, which is a split conformal prediction method that aims to reach equal coverage rates across groups. Experimental results show that our method leads to better EOC than GCQR (see Tables 1-3).
>
> **Q6**: Small max coverage gap of CFQR.
>
> **A6**: We believe that CFQR has better performance related to data distribution. The Column *Max Coverage Gap* in Tables 1 to 4 is defined as the average maximum difference in coverage rates between groups for all bins in L251. For Adult data, the challenging group (A=1) mainly falls in the first few bins. CFQR only has a few bins showing large coverage gaps and the gaps in other bins are extremely small, leading to a small average. For example, when we set random seed as $1$ and run CFQR, each group has a coverage rate of 0, 0.6, 1.2 in bin 1 and 98.28, 99.15, 99.45 in bin 2, while the other 8 bins either are fully covered or have no samples. Therefore, the mean of max coverage gaps is 0.24 while group A=1 has a conditional coverage rate of  74.97, and groups A=0 and A=2 have 92.52 and 94.86, respectively. We observe a similar phenomenon for *MEPS* data in Table 3, where CFQR has a large gap in the first bin and small gaps for other bins. We will explore more advanced binning strategies in future works.

---

> > ### Comment · Reviewer_RXWV · 2023-08-20
> >
> > I thank the authors for their detailed response. After reading the rebuttal and the uploaded pdf my remaining question is regarding the new added figure 4. Why does it seems that LCQR (figure e) provides  better coverage across bins and groups than the proposed BFQR (figure f)? LCQR seems to have a coverage closer to 0.9 across all bins while BFQR seem to have larger coverage disparities across different bins (e.g., bin 1 coverage is around ~0.65 and for bin 4 its 1.0) It would be good to discuss this to understand why using BFQR instead of LCQR would be beneficial. Thank you.

---

> > > ### Author Response · Authors · 2023-08-20
> > > **Comparison between LCQR and BFQR**
> > >
> > > We would like to provide further clarification regarding that LCQR is designed for classification problems. We adapt it to the regression setting by treating samples within the same bin as a class. Notably, both LCQR and BFQR employ the same binning methodology. However, distinctions between the two emerge as follows: LCQR uniformly assigns coverage rates to all bins, ensuring an overall coverage of 0.9. In contrast, BFQR accommodates differential coverage rates across bins, as long as their cumulative mean stands at 0.9. This conforms to the requirement of EOC that coverage rates for different groups with similar outcomes are close.
> > >
> > > LCQR suffers from increased interval widths because it enforces bins with large uncertainty (e.g., bins 1 \& 10) to have coverage rates as high as other bins. As shown in Table 2 in our paper, the mean width of LCQR is around 160,108. In contrast, BFQR is able to adjust bin coverage rates based on the uncertainty of each bin through the method proposed in Section 4.3, leading to a mean width of only 91,970. Experiment results in Tables 1-3 also support that LCQR is inferior to BFQR both in EOC and interval width. Therefore, we believe BFQR is more beneficial than LCQR.

---

### Official Review · Reviewer_219P · 2023-07-07

**Soundness:** 3 good
**Presentation:** 3 good
**Contribution:** 3 good
**Rating:** 6
**Confidence:** 3

**Summary:**

This paper studies the problem of algorithmic fairness using uncertainty quantification methods in regression settings. The main novelty of this work over previous work is taking the notion of “equalized coverage” and strengthening it by considering finer groups by additionally conditioning on the target variable. The result is a conformal prediction based post-processing method that provably gives conditional coverage guarantees over sets defined by the demographic groups and target variable, where most prior work just considers sets on the group membership. Another upshot of this method is that the conformal-based post-processing method proposed (BFQR) explicitly focuses on also optimizing interval widths of predictions to improve “the amount of decision-making information available” to downstream decision-makers.

The paper is organized by presenting the main fairness notion studied - equal opportunity of coverage (EOC), describing the conformal prediction based post-processing approach to improve EOC and optimize for prediction interval widths. The main theoretical results are marginal and conditional coverage guarantees of BFQR. Finally, there is experimental evaluation on one synthetic and two benchmark datasets with comparisons to a few other conformal prediction based regression procedures.


**Strengths:**

This paper studies an important problem - giving conditional coverage guarantees, while conditioning on the true label of the data, using a general post-processing conformal method, with particular emphasis on minimizing interval width to give more utility to downstream decision makers. The presentation is clear and precise.

**Weaknesses:**

A main upshot of BFQR as presented is the improvement on interval width, which seems to be modest on 2/3 (synthetic and Adult) datasets. It seems hard to compare this statistic with the other methods and evaluate how it compares to BFQR’s slight degradation in tightness of coverage rates.


**Questions:**

There are a few things that would be helpful to clarify in your experimental setup. What are the groups / conditioning events for the comparison experiments run with GCQR and MVP? My understanding is that the sets they give conditional guarantees for do not need need to condition on the true label - but can use them if given the information. If tested in this way, could it take some of the bite out of the improvement we see in EOC with BFQR / is that already how the experiments were run?


**Limitations:**

-

---

> ### Author Rebuttal · Authors · 2023-08-10
>
> We extend our sincere appreciation for your valuable feedback and suggestions. Regarding your concerns, we would like to offer further clarification.
>
> **Q1**: Modest improvement in interval width and degradation in coverage rates.
>
> **A1**: The research problem of the trade-off between interval width and coverage rate is significant. That is, with the same samples and calibration method, when we set a high coverage rate, the width of the prediction interval tends to increase. Therefore, our goal is to achieve EOC (the desired coverage rate for different demographic groups) meanwhile reduce the prediction intervals, not to maximize coverage rate. The width of the prediction interval can be also influenced by other factors:
>
> i) uncertainty of the sample: the hardness of predicting the distribution of outcomes for the base model significantly contributes to the lower boundary of the prediction interval width.
>
> ii) proper quantification and calibration: if the uncertainty is not properly quantified and adjusted, we may not achieve the desired coverage rate or at a cost of wider intervals.
>
> For *Adult* and *MEPS* data, our proposed BFQR shows a significantly lower interval width than baselines with similar coverage rates of around 0.9. BFQR* is a variation of BFQR that fills the noncontinuous space between prediction intervals of BFQR (Lines 270-272). For example, if BFQR gets intervals of [0,1], [4,5], and [10,15], then the output of BFQR* would be [0,15]. Enforcing continuity in BFQR* inevitably results in larger widths and a higher coverage rate than BFQR.
>
> For synthetic data, the improvement of BFQR in width is modest compared with CQR, while significant compared with other methods. We believe it is because we utilize the result of CQR as the starting point, and the greedy strategy in optimization may lead to early stopping. However, as presented in Table 4, other optimization methods exhibit inferior results w.r.t. the width, running time, and other metrics. In future work, to get narrower widths, we will advance our method by improving the calibration process (factor ii).
>
> **Q2**: Unclear experimental settings.
>
> **A2**: We would like to further clarify our experimental settings. (1) No true label is given in the test stage and we only use true labels for evaluation. (2) GCQR, MVP, and CFQR are only conditioned on sensitive attributes, i.e., they focus on equalized coverage. BFQR aims to achieve EOC which is conditioned on both sensitive attributes and true labels.

---

### Author Rebuttal · Authors · 2023-08-10

We sincerely appreciate all the recognition and suggestions from all reviewers, which greatly contribute to enhancing both the quality and clarity of our paper. We have thoughtfully addressed several common concerns raised by the reviewers and would incorporate them into the next version of our paper:

1. To illustrate the computation of $\beta_m$ and the corresponding quantile values in Lines 161-171, we have added Figure 3 to the attached PDF, along with a detailed explanation.

2. In an effort to provide a clear understanding of our work, we have included rigorous definitions for the key concepts, such as $p$, $p'$, and $d_{TV}(p,p')$ in Lines 116-122.

3. Recognizing the importance of clarity in our presentation of experiments, we have further elaborated on the experimental settings and results in our responses. Additionally, we have introduced Figure 4 in the attached PDF to offer enhanced visual comparisons between the different methods.

---

### Decision · Program_Chairs · 2023-09-21

**Decision:**

Accept (poster)

**Comment:**

The paper proposes an equal opportunity analogue of "equalized coverage", which they call "equal opportunity of coverage". It is a two-fold objective that tries to have parity for coverage rates for different groups with similar outcomes and the overall coverage rate for the entire population is lower bounded. The paper proposes Binned Fair Quantile Regression, a post-processing method to improve EOC which is agnostic to the model based on calibrating on a held-out set and uses conformal prediction and constrained optimization to optimize for the bin width.

The reviewers agreed the paper presents some nice and well-motivated ideas with promising results. One reviewer raised concerns around the novelty referencing previous works related to distribution matching post-processing techniques in fair regression; however it appears they are not directly applicable here since the present work involves conformal interval prediction. During discussions, the only concern raised was that the proposed method requires some trade-offs in some metrics such as the original equalized coverage. I believe this is reasonable as EOC is a different goal from equalized coverage and other metrics; moreover, the paper highlights this and provides practical insights around this. Overall, this paper is a worthwhile contribution.